



# Spatial and temporal variability of snowfall over Greenland from CloudSat observations

Ralf Bennartz[1,2], Frank Fell[3], Claire Pettersen[2], Matthew D. Shupe[4], Dirk Schuettemeyer[5]

[1]Earth and Environmental Sciences Department, Vanderbilt University, Nashville, Tennessee, USA
[2]Space Science and Engineering Center, University of Wisconsin - Madison, Madison, Wisconsin, USA
[3]Informus GmbH, Berlin, Germany
[4]Physical Sciences Division, Cooperative Institute for Research in Environmental Science and NOAA Earth System
Research Laboratory, Boulder, Colorado, USA
[5]ESA-ESTEC, Nordwijk, The Netherlands

*Correspondence to*: Ralf Bennartz (ralf.bennartz@vanderbilt.edu)

**Abstract.**

We use the CloudSat data record 2006 - 2016 to estimate snowfall over the Greenland Ice Sheet (GrIS). We first evaluate CloudSat snowfall retrievals with respect to remaining ground-clutter issues. Comparing CloudSat observations to the GrIS

topography (obtained from airborne altimetry measurements during IceBridge) we find that at the edges of the GrIS spurious high snowfall retrievals caused by ground clutter occasionally affect the operational snowfall product. After correcting for this effect, the height of the lowest valid CloudSat observation is about 1200 meters above the local topography as defined by IceBridge. We then use ground-based Millimeter Wavelength Cloud Radar (MMCR) observations obtained from the Integrated Characterization of Energy, Clouds, Atmospheric state, and Precipitation at Summit, Greenland (ICECAPS) experiment to

devise a simple, empirical correction to account for precipitation processes occurring between the height of the observed CloudSat reflectivities and the snowfall near the surface. Using the height-corrected, clutter-cleared CloudSat reflectivities we next evaluate various Z-S relationships in terms of snowfall accumulation at Summit through comparison with weekly stake field observations of snow accumulation available since 2007. Using a set of three Z-S relationships that best agree with the observed accumulation at Summit, we then calculate the annual cycle snowfall over the entire GrIS as well as over different

drainage areas and compare the so-derived mean values and annual cycles of snowfall to ERA-Interim reanalysis. We find the annual mean snowfall over the GrIS inferred from CloudSat to be 34 ± 7.5 cm/yr liquid equivalent (where the uncertainty is determined by the range in values between the three different Z-S relationships used). In comparison, the ERA-Interim reanalysis product only yields 30 cm/yr liquid equivalent snowfall, where the majority of the underestimation in the reanalysis appears to occur in the summer months over the higher GrIS and appears to be related to shallow precipitation events.

Comparing all available estimates of snowfall accumulation at Summit Station, we find the annually averaged liquid equivalent snowfall from the stake field to be between 20 and 24 cm/yr, depending on the assumed snowpack density, and from CloudSat



23 ± 4.5 cm/yr. The annual cycle at Summit is generally similar between all data sources, with the exception of ERA-Interim reanalysis, which shows the aforementioned under-estimation during summer months.

**1 Introduction**

The Greenland Ice Sheet (GrIS) is currently losing mass at a rate of roughly 240 Gt/yr, translating into a sea level rise of

0.47±0.23 mm/yr (van den Broeke et al., 2016), which corresponds to roughly 15-20% of the total annual mean sea level rise. Precipitation is the sole source of mass of the GrIS. Additionally, the inter-annual precipitation variability appears to be the main driver of inter-annual variability in the mass balance of the GrIS and the largest source of uncertainty in the surface mass balance of the GrIS (van den Broeke et al., 2009). At the same time, ground-based long-term observations of precipitation over the GrIS are sparse. Over the GrIS surface station networks include the Greenland Climate Network (GC-NET, Box and

Steffen (2000)) and the 'Programme for Monitoring of the Greenland Ice Sheet' (PROMICE; by the Danish Energy Agency (van As, 2017)). These networks of surface stations do not directly observe precipitation, however limited information on accumulation may be inferred by boom height measurements using sonic ranging. van den Broeke et al. (2016) point out the importance of individual mass sources, i.e. snowfall, contributing to the GrIS surface mass balance which '[…] must be interpolated from scarce in-situ measurements and/or simulated using dedicated regional climate models, which introduces

potentially large uncertainties'. Comparison studies of different climate and reanalysis models show an about 25% - 40% spread in snowfall estimates over the GrIS between the different models (Cullather et al., 2014; Vernon et al., 2013).

Since 2006, CloudSat observations have been used in a series of studies addressing snowfall globally (Liu, 2008; Hiley et al., 2011; Palerme et al., 2014; Kulie et al., 2016; Adhikari et al., 2018; Kulie and Milani, 2018). While some of the global studies include Greenland, a detailed assessment of snowfall over the GrIS using CloudSat has to our knowledge not yet been

performed. Here we present such an assessment.

Several factors complicate snowfall estimates from CloudSat over the GrIS. First, the height of the central GrIS makes for a unique environment in terms of snowfall characteristics. For example, Pettersen et al. (2018) find that a large fraction of the total accumulation at Summit Station falls from clouds that contain no liquid water with only ice microphysical processes involved. Secondly, the steep altitude changes at the edges of the ice sheet pose challenges to the space-borne radar

observations due to ground clutter. This issue has a compounding impact on CloudSat's clutter-affected blind zone near the surface, which typically extends from the surface to about 1200 m (Maahn et al., 2014). Thus, 'surface' snowfall rates observed from CloudSat are typically taken at an altitude of 1200 m above the surface.

To account for the above issues related to surface topography and CloudSat's blind zone, we use auxiliary information about the GrIS surface topography (the IceBridge BedMachine V3 topography, Morlighem et al. (2017)) to better characterize which

CloudSat radar bin can be regarded as the lowest clutter-free observation. We then use Millimeter Wave Cloud Radar (MMCR) observations from ICECAPS to estimate the impact of CloudSat's observation height on estimated 'surface' snowfall, and propose a simple GrIS-specific empirical correction to account for the difference between actual surface snowfall and the



CloudSat observations made at about 1200 m height above the surface. We further use ground-based snowfall accumulation from Summit Station to assess the validity of different Z-S relationships applied to CloudSat. Based on these corrections and findings, we then calculate our best estimates of surface snowfall from all available CloudSat observations and assess the annual cycle and spatial distribution of snowfall over the entire GrIS.

5     The remainder of this paper is structured as follows. In Section 2 we discuss the different data source including the ground-based observations at Summit Station, CloudSat satellite observations, and reanalysis products. Section 3 addresses the aforementioned issues related to using CloudSat as a proxy for surface snowfall. In Section 4 we compare CloudSat and ERA-Interim reanalysis data products and provide estimates for the annual cycle of snowfall over the different drainage systems of the GrIS as well as over Summit. Conclusions and an outlook are provided in Section 5.

## 10   2 Datasets and Methods

### 2.1 CloudSat data

CloudSat (Stephens et al., 2002; Stephens et al., 2008) carries the single-frequency, W-band (94 GHz) Cloud Profiling Radar (CPR; Tanelli et al., 2008). CPR has provided global cloud and precipitation profiles since 2006, however, only daytime scenes can be observed since 2011due to a hardware failure[1]. The CPR is a non-scanning, near-nadir pointing instrument with a mean

15   spatial resolution of ~1.5 km and a vertical range gate spacing of 500 m, although instrument oversampling enables 240 m data bins in the CloudSat data products. In the framework of this study, we use the 2C-SNOW-PROFILE (Wood et al., 2014) together with the GEOPROF reflectivity profiles and ECMWF-AUX temperature and moisture profiles (Stephens et al., 2008). Product documentation can be obtained from the CloudSat Data Processing Center (DPC, http://www.cloudsat.cira.colostate.edu/). All analysis is based on the CloudSat Release 5 data, which were made publicly

20   available by the DPC in June 2018.

Figure 1 shows the number of CloudSat measurements available over Greenland (top) and within a 50 km range from Summit Station (bottom). One can clearly identify the reduction in data coverage after the CloudSat battery failure in April 2011. Even after operations were restored, data collection was limited to the sunlit part of the orbit, leading to a slight annual cycle in the number of observations available over Greenland.

---

[1]  For a detailed timeline of CloudSat availability, see the operational CloudSat Radar Status blog at: https://cloudsat.atmos.colostate.edu/news/CloudSat_status





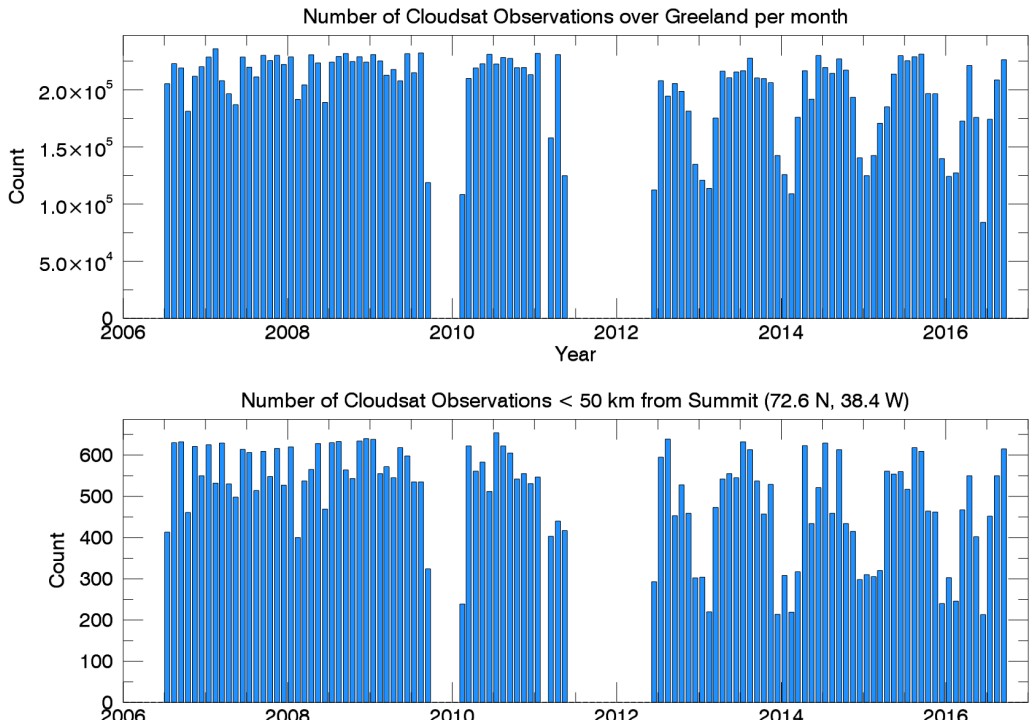

**Figure 1: Number of CloudSat observations per month all over Greenland (top) and within 50 km of Summit Station (bottom).**

Figure 2 shows the spatial distribution of these measurements over Greenland. Because of CloudSat's 16-day repeat pattern, coverage at high spatial resolution creates a diamond-shaped pattern over Greenland, as can be seen in the right panel of Figure 2. This pattern limits the maximum resolution of any climatology based on CloudSat data. For a resolution of roughly 100 x 100 km, the coverage appears fairly uniform apart from a north-south gradient, which is a result of by the better coverage near

5    the maximum coverage latitude around 81.8 degrees, due to CloudSat's inclination of about 98.2 degrees.



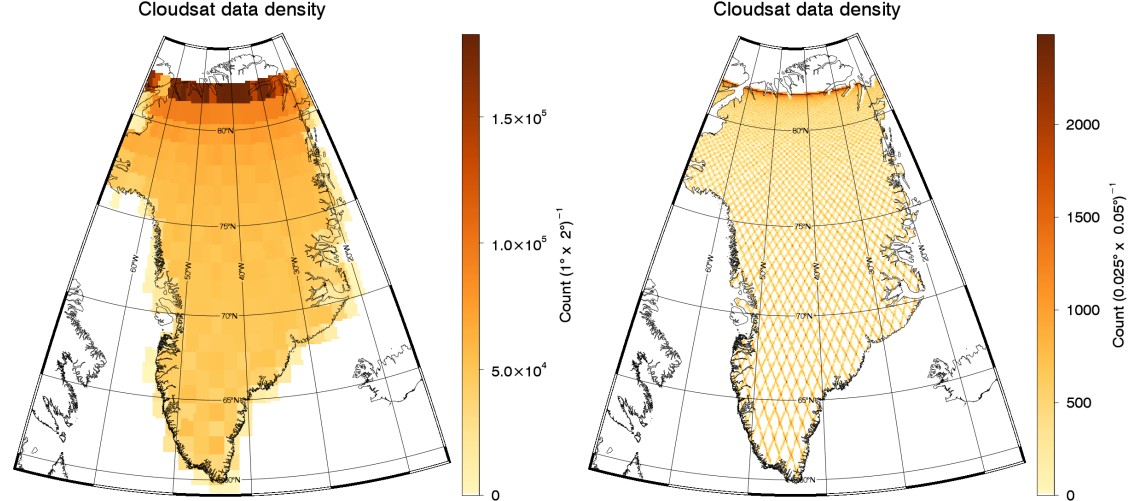

**Figure 2: CloudSat data density over Greenland at 1x2 degrees (left, approximately 100 km x 100 km) and 0.025 x 0.05 degrees (right, approximately 2.5 km x 2.5 km). Shown is the total number of observations per grid box over the time period from 2006 to 2016.**

### 2.2 ICECAPS observations

The ICECAPS (Integrated Characterization of Energy, Clouds, Atmospheric state, and Precipitation at Summit) experiment

operates a sophisticated suite of instruments that observe properties of the atmosphere, clouds, and precipitation at Summit Station, located near the apex of the GrIS at an elevation of 3200 m above sea level (Shupe et al., 2013). These instruments have been in operation at the NSF Mobile Science Facility nearly continuously since July 2010. This comprehensive dataset of atmospheric properties above the GrIS is unprecedented, due to both the large number of distinct and complementary measurements that are being made and the length of the dataset.

The ICECAPS experiment, as well as instrument specifications, measurements, and derived products, are described in Shupe et al. (2013). In particular, the Ka-band MMCR, the X-band Precipitation Occurrence Sensor System (POSS), and the Multi-Angle Snowflake Camera (MASC, 2015 – 2016, selected dates only) are of relevance with respect to precipitation.  In this study we build on earlier work by Castellani et al. (2015) and Pettersen et al. (2018). The latter study also provides a combined dataset of relevant parameters for studying precipitation variability over the central GrIS. This dataset is available with a

temporal resolution of one minute and is used as a basis for the investigations performed here, but complemented with



additional MMCR observations. For more details on the MMCR and its calibration, see Castellani et al. (2015) and references therein.

In addition to these ICECAPS data, other observations are available at Summit Station. Most important in this context is the so-called 'stake field', which consists of 11 x 11 bamboo stakes planted in a square a few hundred meters away from the station. The height of all 121 stakes above the snow surface is read off approximately every week, thereby creating a unique reference for surface height changes. Once a year, the stakes are raised by about 70 cm in order to allow for continuous measurements. The stake measurements go back to 2003, with easily accessible data going back to 2007, cover the full ICECAPS period, and provide an independent set of observations of snowfall accumulation. Similar to Castellani et al. (2015), who provide a discussion on the accuracy of the stake field data, we herein use these stake observations to assess the other accumulation measurements, namely from the ground-based and spaceborne radar.

**2.3 ERA-Interim**

ERA-Interim (Dee et al., 2011) is a third generation global atmospheric reanalysis provided by the European Centre for Medium-Range Weather Forecasts (ECMWF). It expands on previous versions (e.g. ERA-40) by using an improved atmospheric model and assimilation system (ECMWF, 4D-VAR, 2006). The spatial resolution of the data set is approximately 80 km (T255 spectral) on 60 vertical levels from the surface up to 0.1 hPa. Among other data, it assimilates a long list of satellite radiances (e.g. SSM/I and SSMIS). ERA-Interim data are available since 1979 and are continuously updated. For this study, monthly gridded snowfall estimates were used (downloaded from https://rda.ucar.edu/datasets/ds627.0/).

**2.4 Z-S relations used in this study**

To compare CloudSat CPR and the MMCR, we apply a set of different published Z-S relationships as well as Z-S relationships empirically fitted to some of the observational data. The Z-S relationships used are summarized in Table 1. In addition to the Matrosov (2007, hereafter M07) relationship used in various previous studies, we employ a set of Z-S relationships that apply to single habits, which, based on the above considerations and observations, might be better proxies for snowfall at Summit than M07. While some Z-S relationships are only available at Ka-band (i.e. MMCR), others are available only at W-band (i.e. CloudSat). Only four Z-S relationships are available consistently for both Ka- and W-band and, therefore, allow transferring results between the two bands. Uncertainties of instantaneous snowfall retrievals using Z-S relationships can be large and are discussed in detail in the publications referenced in the last column of Table 1. The individual uncertainties should not be confused with systematic errors introduced by the choice of particular Z-S relationships. Such systematic errors are of greater importance when generating climatological precipitation estimates (as random uncertainties average out in the process of generating e.g. monthly means). Here we follow an approach developed in our earlier publications (Kulie and Bennartz, 2009; Hiley et al., 2011), where we define uncertainties on the climatological end-product in terms of differences between different Z-S relationships. We believe that this approach provides more realistic error estimates on climatological products (for more details, see Hiley et al., (2011))0.



Table 1: Parameters of Ka-band (MMCR) Z-S relations and W-band (CloudSat) used in this study. The POSS operates at X-band so that the Z-S relation is not directly comparable the Z-S relations for MMCR. A Z-S relationship is defined as $Z = AS^B$ where S is the snowfall rate (in mm/hr) and Z radar reflectivity (in mm⁶/m³).

| Name | A (Ka-band) | B (Ka-band) | A (W-band) | B (W-band) | Reference |
|---|---|---|---|---|---|
| M07 | 56.0 | 1.20 | 10.0 | 0.80 | (Castellani et al., 2015; Matrosov, 2007; Pettersen et al., 2018) |
| KB09_LR3 | 24.0 | 1.51 | 13.2 | 1.40 | (Kulie and Bennartz, 2009) using (Liu, 2008) 3-bullet rosettes |
| KB09_HA | 313.3 | 1.85 | 56.4 | 1.52 | (Kulie and Bennartz, 2009) using (Hong, 2007) aggregates |
| L08 | --- | --- | 11.5 | 1.25 | (Liu, 2008) |
| HI11_L | --- | --- | 7.6 | 1.30 | (Hiley et al., 2011) |
| HI11_A | --- | --- | 21.6 | 1.20 | (Hiley et al., 2011) |
| HI11_H | --- | --- | 61.2 | 1.10 | (Hiley et al., 2011) |
| POSS | N/A | N/A | --- | --- | (Pettersen et al., 2018; Sheppard and Joe, 2008) |

**2.5. Assumption on snowpack density used in this study**

Dibb and Fahnestock (2004), based on snow pits, find snow densities over central Greenland between 240 and 370 kg/m³ at depths between 15 - 45 cm, which is probably a realistic average depth range for annual accumulation studies. More recent results by Fausto et al. (2018) indicate an average density of the uppermost snow layer over Greenland to be 315±44 kg/m³

10   for the uppermost 10 cm, and 341±37 kg/m³ for the uppermost 50 cm. Fausto et al. (2018), as well as earlier studies (e.g. Reeh et al. (2005)), also find a weak dependency of density on temperatures. Here we use snowpack density in two ways: Firstly, in Section 3.4.2 we use the stake field observations as a reference to assess the validity of different CloudSat-based Z-S relationships. Following the work of Castellani et al. (2015), we compared the stake field data with CloudSat-based liquid equivalent snowfall rates by calculating the effective density needed for the CloudSat snowfall estimates to match the observed

15   accumulation. We then reject as unrealistic Z-S relationships that fall outside the above wide range of densities reported for Summit. Secondly, in Section 4.2 we compare the annual cycle of liquid equivalent snowfall over Summit from different observations. To convert the stake field values into liquid equivalent snowfall we use the conversions proposed by Fausto et al. (2018) and Reeh et al. (2005) to provide a range of liquid equivalent snowfall rates based on the stake field observations.



### 3 Evaluation of CloudSat observations

Here we assess the full CloudSat snowfall dataset over Greenland in terms of its viability for climatological snowfall studies. Amount and spatial distribution of the data used for this assessment are shown in Figure 1 and Figure 2. We analyse the dataset with respect to the following issues:

    a)   Effects and removal of ground clutter.
    b)   Impact of height of CloudSat observation above the surface.
    c)   Impact of choice of Z-S relationship.

At Summit, concurrent observations from stake field and MMCR allow for a detailed assessment of these issues.

### 3.1 Effects and removal of ground clutter

CloudSat observations in the lowest range bins above the surface are affected by ground clutter. Because of topography, this effect is more pronounced over land than over ocean. The CloudSat SNOWPROF product accounts for the impact of ground clutter by providing a confidence flag for the retrieved surface snowfall rates. This flag depends on the type of surface as well as on other criteria, such as vertical consistency of retrieved snowfall rates. A key input over the highly structured coastal terrain of Greenland and the edges of the GrIS is the height of the surface bin, which describes where the radar beam first

interacts with the surface. This quantity is provided in the CloudSat data and is retrieved from the radar reflectivity itself, as well as from an underlying digital elevation model.

In our analysis, we found that the height of the surface bin is not always accurately represented over Greenland. This occasionally causes significant outliers in the retrieved surface snowfall rate. In order to study and possibly correct for this issue, we use the IceBridge BedMachine (V3) surface topography measurements (Morlighem et al., 2017) and collocate those

with each individual CloudSat observation. We then re-derive the snowfall rates based on the 5th radar bin above the surface as defined by this new topography. We compare these new retrievals to the originally retrieved snowfall rates, which also typically are taken from the 5th radar bin above the surface, but with a different prescribed surface elevation model defining the surface. We restrict our analysis to SNOWPROF confidence flag values 3 and 4,which indicate high confidence in the retrieval.

The difference in elevation reported between CloudSat and BedMachine can be seen in Figure 3. It is not entirely unrealistic that some of the differences seen between the two digital elevation models are caused by melting in the ablation zone. However, differences might also be caused by other factors. Clearly, some of the coastal regions with the largest differences experience significant amounts of snowfall. The effect of using different underlying surface topographies for snowfall retrieval and accumulation is shown in Figure 4 through Figure 6.

Figure 4 shows two-dimensional histograms of radar reflectivity and derived SNOWPROF surface snowfall rate. For each reported snowfall rate, the corresponding radar reflectivity was obtained from the CloudSat GEOPROF product. The left panel of Figure 4 shows the surface snowfall rate and reflectivity reported directly from the product. The right panel shows the



revised snowfall rate accounting for the BedMachine topography. Note that for the right panel all snowfall rates are also directly retrieved from SNOWPROF. In contrast to the left panel, the snowfall rates are occasionally taken from radar bins higher in the atmosphere to account for the higher topography estimates from BedMachine.

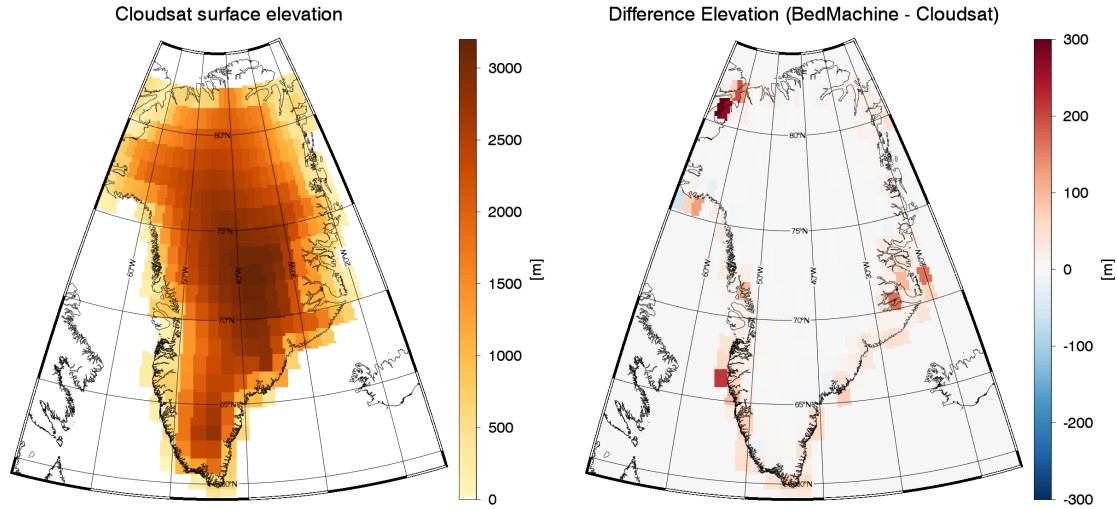

**Figure 3: The left panel shows the mean elevation reported from CloudSat binned to 1 x 2 degrees. The right panel shows the difference between the elevation difference between IceBridge BedMachine v3 (Morlighem et al., 2017) and CloudSat. Note, that open water and sea ice observations are excluded from the dataset, so that differences observed near the coast only stem from ice-free land or GrIS observations within each grid box.**

5   Comparing the panels in Figure 4, we note that there is a significant number of high reflectivities associated with very high and often physically implausible snowfall rates of up to 50 mm/h (see upper right part of left panel of Figure 4). Using the BedMachine topography eliminates these high snowfall rates (see right panel of Figure 4). A visual inspection of a few of these cases also indicates clutter-affected observations in the original CloudSat product, which are successfully eliminated using the BedMachine topography. The revised formulation for the lowest valid CloudSat bin above the surface thus leads to

10   a significant reduction surface clutter as shown in Figure 4.





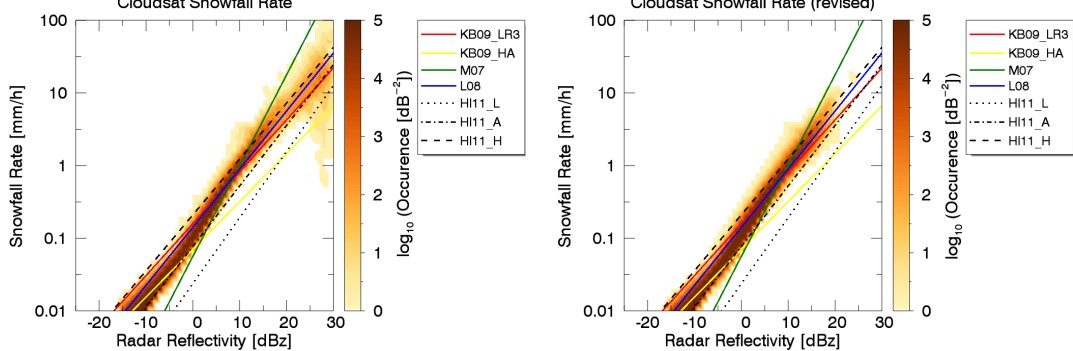

**Figure 4: Histogram of occurrence of snowfall rate versus radar reflectivity for the entire CloudSat dataset over Greenland. The left panel shows the relation if the SNOWPROF surface snowfall rate is used face value. The right panel shows the relation with corrected topography. Different Z-S relations are shown as well.**

The effect of the above revisions can be seen in Figure 5, which shows the actual heights of CloudSat "surface snowrate" observations above the surface, as well as the difference between the original SNOWPROF heights and the revised heights. One can see that in the original formulation the distance to the surface near the coast is often in the 1000 m range, which would likely lead to ground clutter (Maahn et al., 2014), in particular given the complex orography. Note that Figure 5 shows the

effect of the correction on the height of the lowest valid CloudSat observation above the surface (whereas Figure 3 only shows the difference between two topographies). The impact of these differences in topographies (see Figure 3) is amplified as CloudSat observations are binned at 240 m vertical resolution. Because of this 240-m-binning, the slight changes in height observed in Figure 3 can result in larger changes in CloudSat observation height as observed in Figure 5 (e.g. a difference in topography of 50 m might lead to the lowest valid CloudSat bin increasing by 240 m).

Over the central GrIS the average observation height is not affected. We have examined this for the 1x2 degree grid box around Summit Station, where typically the 5th radar bin above the surface is selected (around 1200 m above the surface). In general, it is important to bear in mind that the CloudSat "surface snowrate" observations over structured terrain typically come from about 1200 meters above the surface, and thus do not observe precipitation processes below that altitude. The impact of CloudSat's 'blind zone' below roughly 1200 m above ground for the high GrIS is studied in Section 3.2.

Figure 6 shows the integrated effect of the ground clutter artefacts in the CloudSat surface snowfall rates on accumulation. The mean snowfall rate for all CloudSat data over Greenland would be approximately 0.225 mm/h in the revised formulation. Not correcting for artefacts reduces the mean by 15 % to about 0.2 mm/h, but with significant contributions from larger than 20 dBz, which are eliminated when the IceBridge BedMachine surface topography is applied. Observations that are corrected for ground clutter contribute toward the total snowfall at lower reflectivities, thereby increasing retrieved snowfall rates

between +5 and +15 dBz and increasing snowfall rate in this dBz interval. Note that Figure 6 only presents the grand mean of





all snowfall rates. Because of the large differences in surface elevation between CloudSat and BedMachine near the coasts, the impact of artefacts in coastal areas will be much higher when snowfall climatologies are reported. In contrast, these artefacts will not play a major role in the higher elevations of the GrIS.

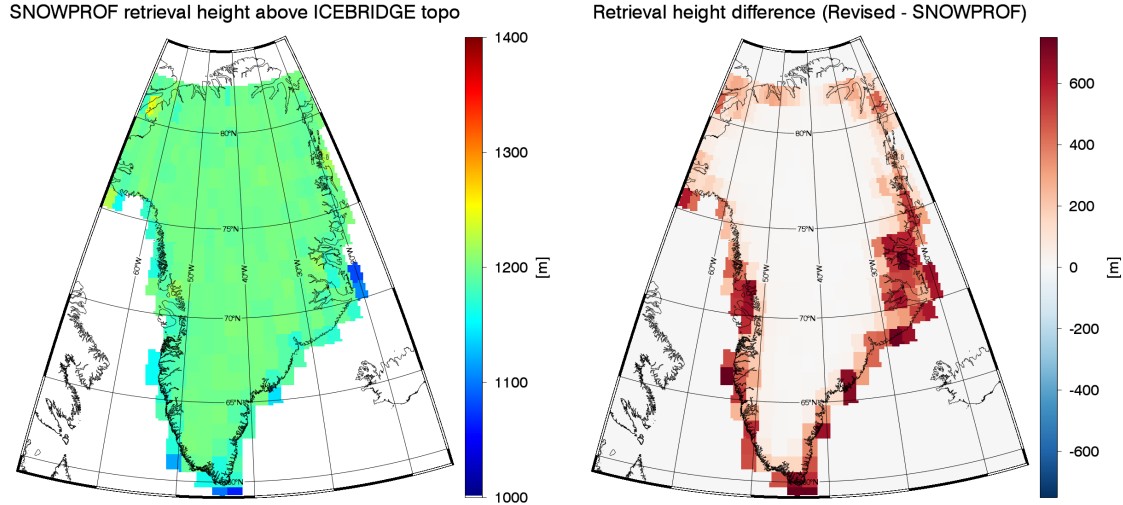

**Figure 5: The left panel shows the height of the CloudSat SNOWPROF 'surface snowrate' observation above the local topography (from IceBridge BedMachine). The right panel shows the difference between the height used in the revised product and original height. For example, at the southern tip of Greenland, the 'surface snowrate' reported in the SNOWPROF product comes from an actual altitude of about 1000 m above the surface. In the revised formulation discussed in the text, this height is pushed up by 500 m to 1500 m.**



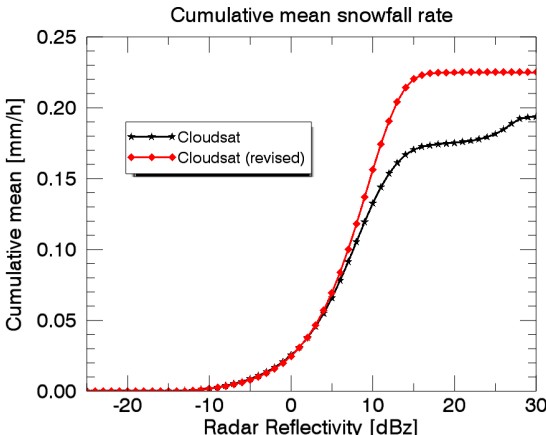

**Figure 6: Impact of surface topography issues on cumulative snowfall rates.**

Based on the results reported in the current section, we will hereon only use the revised snowfall rates that are obtained using the IceBridge BedMachine surface elevation, and discard the surface snowfall rates reported in the SNOWPROF product. We again note that the revised surface snowfall rates we use here are also available in the SNOWPROF product. However,

occasionally, mostly near the coasts, our analysis uses snowfall data from higher radar bins than the SNOWPROF surface snowfall rates to avoid clutter artefacts that would otherwise be present.

**3.2 Impact of height of observations above ground on estimated surface snowfall rate**

As shown in Figure 5, CloudSat snowfall observations over Greenland stem from altitudes of around 1200 m above the surface

to avoid ground clutter issues. This might cause several issues because any precipitation processes happening at lower altitudes are not observed and, consequently, not accounted for in CloudSat estimates.

Here we use MMCR observations from Summit to study the difference between the observed reflectivities at an altitude of 1200 m and those closer to the surface (135 m). We first average the vertical reflectivity profile of the MMCR between 1000 m and 1500 m to account for the vertical resolution of CloudSat. After converting the averaged reflectivity back into dBz units,

we compare it with the MMCR reflectivity observed at 135 m above the surface. This comparison is shown in the left panel of Figure 7. One can see that for most cases, the MMCR reflectivity observed at CloudSat height (1200 m) is lower than the reflectivity near the surface, possibly owed to precipitation processes occurring at altitudes below 1000 m. There are also cases



where the upper reflectivity is higher than the reflectivity near the surface. Cases for such events could include non-precipitating clouds around 1200 m or ice particles sublimating before they reach the surface (virga). These cases might also include situations where the lowest MMCR radar bin saturates under high reflectivities (see Castellani et al. (2015)). Our analysis of this saturation effect shows that 0.3 % of the MMCR observations are impacted, with only a vanishing effect on

the MMCR snowfall rates reported here. The correction developed in the following paragraph is also not affected. We therefore ignore this saturation effect.

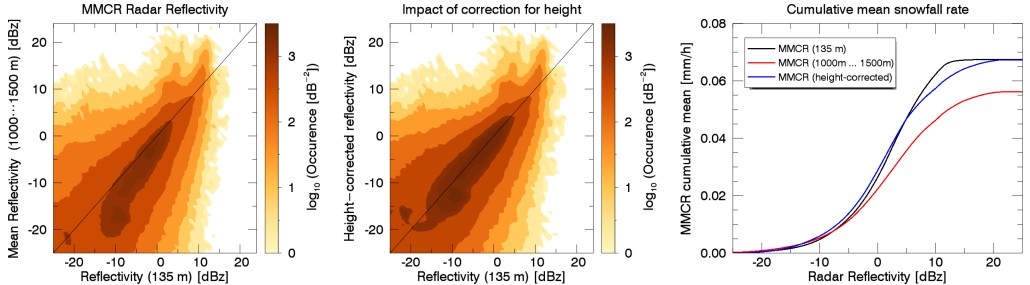

**Figure 7: The two left panels show histograms of MMCR observations at Summit. The left panel compares radar reflectivity at 135 m above the surface with the average radar reflectivity between 1000 and 1500 m above the surface, which corresponds to the height range where CloudSat observations are obtained. The middle panel shows a similar plot but with a height correction applied. See text for details. The right panel shows cumulative mean snowfall rates for the different radar reflectivities show in the left and middle plot.**

By applying the KB09_LR3 Z-S relationship (see Table 1), the red and black curves in the right panel of Figure 7 show the impact of the differences in reflectivity on total cumulative snowfall at Summit. The lower reflectivity at 1000-1500 m yields to an under-estimation of snowfall rate of about 20 % compared to using the reflectivity near the surface. Most of this difference

accumulated in a reflectivity range between -10 and +5 dBz.

In order to correct for this effect, we applied an ad-hoc correction that statistically accounts this effect:

$$dBz_{corrected} = dBz_{1000...1500} + \left[ (1 - 0.2 \cdot dBz_{1000...1500}) > 0 \right] \qquad (1)$$

This statistical correction produces the joint histogram shown in the middle panel of Figure 7 and, by design, matches the total

cumulative snowfall near the surface (see blue curve in right panel of Figure 7). The correction was developed by fitting a regression line to the data between -20 dBz and + 5 dBz. The correction drops to zero at 5 dBz, and thus does not affect reflectivities higher than 5 dBz. While the above formula provides higher corrections for very low reflectivity values, those corrections affect snowfall accumulation only very weakly. For example, for an observed reflectivity of -30 dBz, the correction is +4 dB leading to a corrected reflectivity of -26 dBz, which still does not produce any significant snowfall.

There are caveats to this correction: Importantly, it will only work if the observed atmospheric state is statistically similar to the one on which it was derived. Since the data used for the correction stems from Summit, we expect this correction not to produce viable results outside the high elevations of the GrIS. In order to highlight this limitation, we show in Figure 8 the



same analysis but for Barrow, Alaska. As one can see, the application of the correction outlined in Equation (1) has no effect on the snowfall rate. This is because at Barrow snowfall is produced under different atmospheric conditions. The application of the correction also does not deteriorate the results at Barrow because it has, by design, little to no effect on higher reflectivities. This point is important as near the GrIS ablation zone and in Greenland's coastal regions, one can expect

atmospheric conditions to be more similar to Barrow than to Summit.

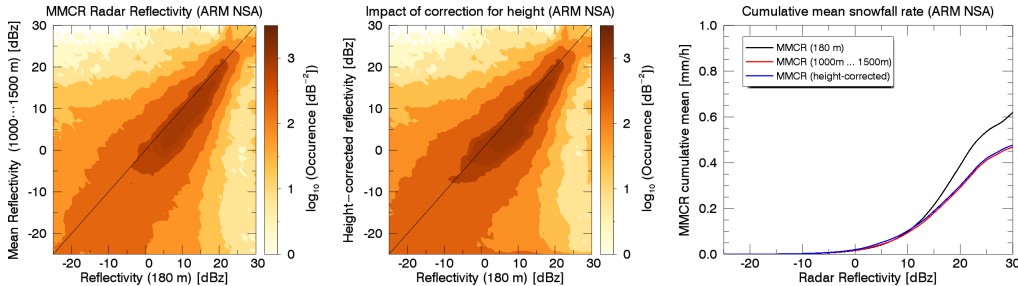

**Figure 8: Same as Figure 7 but for the DOE-ARM site at the North Slope of Alaska (NSA) at Barrow, Alaska. The figures are based on 1.08 million radar profiles obtained between 01/2008 and 4/2011. Only data for the winter months November through April are shown. The Barrow MMCR data were obtained from https://www.archive.arm.gov/discovery/.**

Results presented in Figure 6 and Figure 7 apply to the MMCR, which operates at Ka-band. However, we wish to apply this relationship to CloudSat, which is a W-band radar. Z-S relationships between Ka-band and W-band are different, because medium-sized ice particles enter the Mie-scattering region at a smaller size at W-band than at Ka-band. As snowfall rate increases, the difference between W-band and Ka-band typically increases because the number of large particles outside the

Rayleigh scattering region will increase at W-band. This leads to a different slope of the Z-S relationships at W-band and Ka-band, which might affect the correction proposed here. The Z-S relationships for the two bands are shown in Figure 9 for KB09_LR3. For other Z-S relationships, depending on the ice particles used and, in particular, the underlying size distribution, these differences can be much larger. However, as discussed below, KB09_LR3 is likely more representative of the light snowfall observed over the high GrIS than other Z-S relationships, which apply more to the mid-latitudes. From Figure 9, one

can identify the slight difference in slope between Ka-band and W-band. However, since the above-proposed correction only has significant effect in the range between -10 dBz and +5 dBz, the impact of the earlier onset of Mie-scattering in W-band versus Ka-band will be very small.




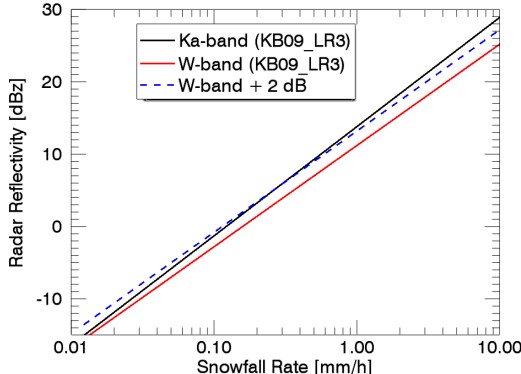

**Figure 9: Radar reflectivity at Ka-band (MMCR) and W-band (CloudSat) as function of snowfall rate for KB09_LR3 Z-S relation. The blue dashed curve has the same slope as the red curve and is shown only as a visual reference to see the slight difference in slope between the red and black curves.**

Based on this discussion, we will apply the above-formulated correction to CloudSat observations without further modification for radar wavelength. This will affect the retrieved snowfall estimates over the higher elevations of the GrIS, but will have little effect on estimates in the ablation zones near the coast, where snowfall is expected to be associated with higher reflectivities. In future studies it would be interesting to look at this issue further and study, for example, potential temperature

dependencies. Initial results from Summit do show a weak dependency of the correction on surface temperature (not shown). We have also tested for a dependency on precipitation type using the classification by Pettersen et al. (2018), but did not find any significant differences in the correction between their IC (ice-only cloud) and LWC (liquid-water containing) clouds. However, expanding this analysis to more Arctic sites, such as Barrow, might allow for a more general correction that would help mitigate some of the issues related to the height above surface of CloudSat snowfall estimates.

A similar correction method has recently been developed by Souverijns et al. (2018). In contrast to our correction, their method works on retrieved snowfall rates rather than reflectivities. Their method is not directly applicable to our approach because we use a set of Z-S relationships to determine uncertainty (see Section 2.4). Thus, a correction based on snowfall rate (rather than reflectivity) would be dependent on which Z-S relationship is used. In general, we prefer performing such corrections on observations (radar reflectivity) rather than retrieved quantities (snowfall rate).

### 3.3 Impact of Z-S relation

Figure 10 shows cumulative snowfall rates based on the full CloudSat dataset in a similar manner as Figure 6. The original CloudSat SNOWPROF optimal estimation retrieval, as well as L08 and KB09_LR3 and M07, are relatively similar in their results (to within ± 10%). All other Z-S relationships fall outside that range. The right panel of Figure 10 shows the impact of

the height-correction (previous section) on the retrieval, which, by example of the 'KB09_LR3' relationship, increases




cumulative snowfall rate near 30% at around 0 dBz and 7% at around 30 dBz (difference between red curves between the left and the right panel). Note that for the original SNOWPROF CloudSat retrieval this correction cannot be applied, as the original retrieval is an optimal estimation retrieval that cannot simply be re-calculated with revised reflectivities.

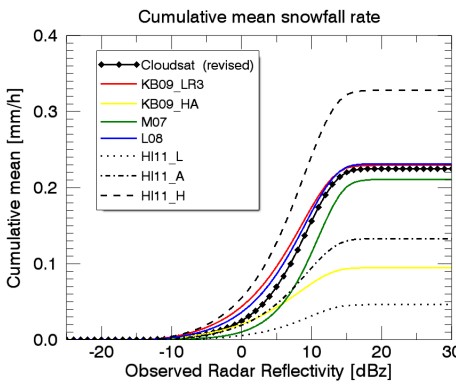
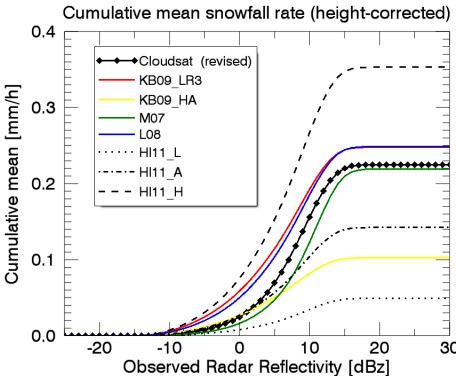

**Figure 10: Cumulative snowfall rates derived from all CloudSat observations over Greenland. The thick black line corresponds to the CloudSat-derived surface snowfall rate from SNOWPROF with ground-clutter removed ('revised', same as in Figure 6). The other lines correspond to the Z-S relations applied to CloudSat reflectivities without height correction (left panel) and with height correction (right panel).**

Figure 10 also highlights the importance of low detectability thresholds for space-borne precipitation radar if GrIS snowfall is to be observed. About 50% of the total accumulation over the GrIS occurs at reflectivities between -10 dBz and +7 dBz. A minimum radar detectability threshold should therefore be lower than -10 dBz to accurately account for snowfall over the GrIS.

### 3.4 Comparison of Summit Station snowfall estimates against stake field

Snow stake field accumulation measurements are available from 2007 onwards and therefore coincide with CloudSat data availability, as well as with the availability of MMCR observations from ICECAPS. Here we compare both radars to the stake field observations.

### 3.4.1. MMCR versus stake field

In Figure 11, we compare liquid equivalent snowfall accumulation derived from MMCR and POSS with the geometric accumulation obtained from the stake field. The ratio between the two quantities is the effective density the snowpack would need to have to explain the accumulation via the radar-derived liquid equivalent snowfall rates. As mentioned earlier, the stake



field is read typically once per week. In order to obtain corresponding MMCR (or POSS) accumulation for each stake field observation period (week), we added up all MMCR (or POSS) snowfall rates for that same period. For the MMCR we used three different Z-S relationships. The POSS reports snowfall rate based on Sheppard and Joe (2008), so only this one value was obtained. Figure 11 shows the so-derived liquid equivalent snowfall rate from MMCR (or POSS) plotted against the stake

5  field accumulation. The ratio between the stake field accumulation and the MMCR (or POSS) accumulation can be interpreted as the effective density that would be needed to explain the observed stake field snowfall accumulation by the liquid equivalent snowfall of the MMCR or POSS (see also Castellani et al., (2015)).

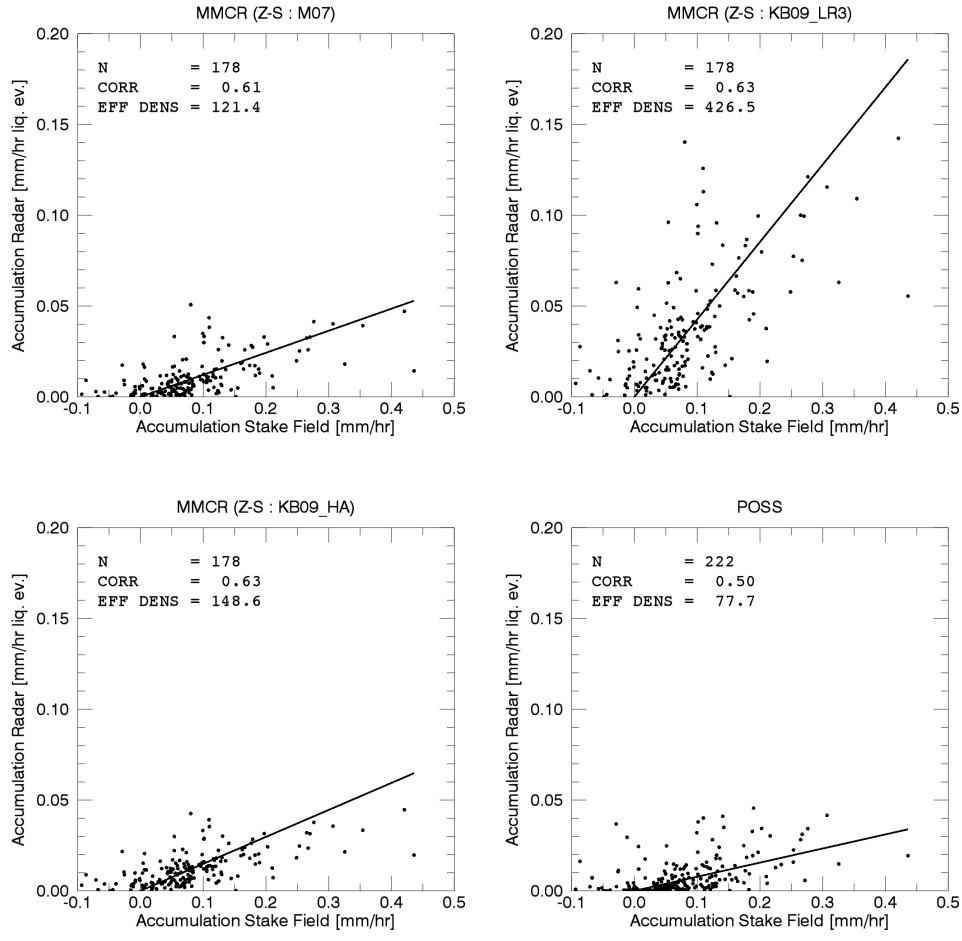

**Figure 11:** Comparison between snowpack accumulation rates from stake field and MMCR-derived or POSS-derived liquid equivalent accumulation rates. 'EFF DENS' is the effective density in [kg/m³] of the snowpack





**needed to explain the mean stake accumulation by the liquid equivalent precipitation from either MMCR or POSS. The lines start at [0,0] and have the reported effective density (divided by 1000) as slope. Each data point corresponds to one week of observations as the snow stake heights are read of typically once a week.**

One can see that between the different Z-S relationships used, most yield an effective snowpack density around 100 kg/m$^3$. Only KB09_LR3 yields a significantly higher effective density of 426 kg/m$^3$. As discussed in Section 2.5, observed densities in the upper snow layers at Summit are likely in the range of 240-380 kg/m$^2$. None of the above Z-S relationships falls into that range. Based on this consideration and the slightly higher correlation between stake field and radar estimates, KB09_LR3

is likely closest to a representative Z-S relationship for Summit, although it likely over-estimates actual accumulation slightly. KB09_LR3 would also be most consistent with the type of snowfall often observed at Summit, that is mostly individual ice crystals with little aggregation or riming (e.g. Pettersen et al., 2018). Deposition onto (or sublimation from) the snow surface is not accounted for in these estimates, which would somewhat enhance (reduce) the accumulation observed by the radar. Based on ERA-Interim reanalysis data, the effect of sublimation/deposition is very small (~ 2%).

### 3.4.2. CloudSat versus stake field

To compare CloudSat observations with the stake field, we selected all CloudSat data (height- and clutter-corrected) within 50 km from Summit and averaged it over the time intervals between stake field observations, which are typically in weekly intervals. We rejected any match-ups where there were less than 30 CloudSat observations within a given stake field time

interval. This resulted in 369 pairs of weekly accumulation statistics from CloudSat and concurrent stake field observations over the time period 2007-2016.

Figure 12 shows the accumulation rates obtained from CloudSat compared to the stake field for those 369 data points for different Z-S relationships. Compared to the corresponding figure for MMCR (previous section), correlations are much lower. This increased scatter is not surprising given that CloudSat provides only one to three orbits per week around Summit.

However, in terms of total accumulation over longer time periods, CloudSat does show a good agreement with the stake field as shown in Figure 13. We note that the good agreement of the total accumulation seen in Figure 13 is by design, as the effective density is used to scale the Cloudsat observations to the stake field. However, the curves follow each other closely over the entire observation period, which could not necessarily be expected if, for example, CloudSat would preferably sample certain types of snowfall. The good agreement in accumulation is despite the large scatter between CloudSat and the stake

field seen in Figure 12. This scatter can partly be explained by CloudSat not being perfectly collocated in space and time with the ground-based observations, and by relatively few individual CloudSat overpasses contributing to each weekly average. Often CloudSat might miss an individual snowfall event and hence report a near-zero snowfall rate. In other cases, CloudSat might observe a single snowfall event, which is not representative for the entire week and thereby over-estimate the weekly snowfall. Both effects can be observed in Figure 12.





Similar to the above discussion on MMCR, choosing an appropriate Z-S relationship remains critical in terms of the effective density needed to transfer CloudSat liquid equivalent snowfall rates to accumulation. The four Z-S relationships shown in Figure 12 provide effective density values between 181 and 365 kg/m$^3$, providing a generally better agreement with the numbers of 240-380 kg/m$^2$ discussed in Section 2.5 The three Z-S relationships ('HI11_H', 'KB09_LR3', and 'L08') produce

5    a mean value of 298 kg/m$^2$.

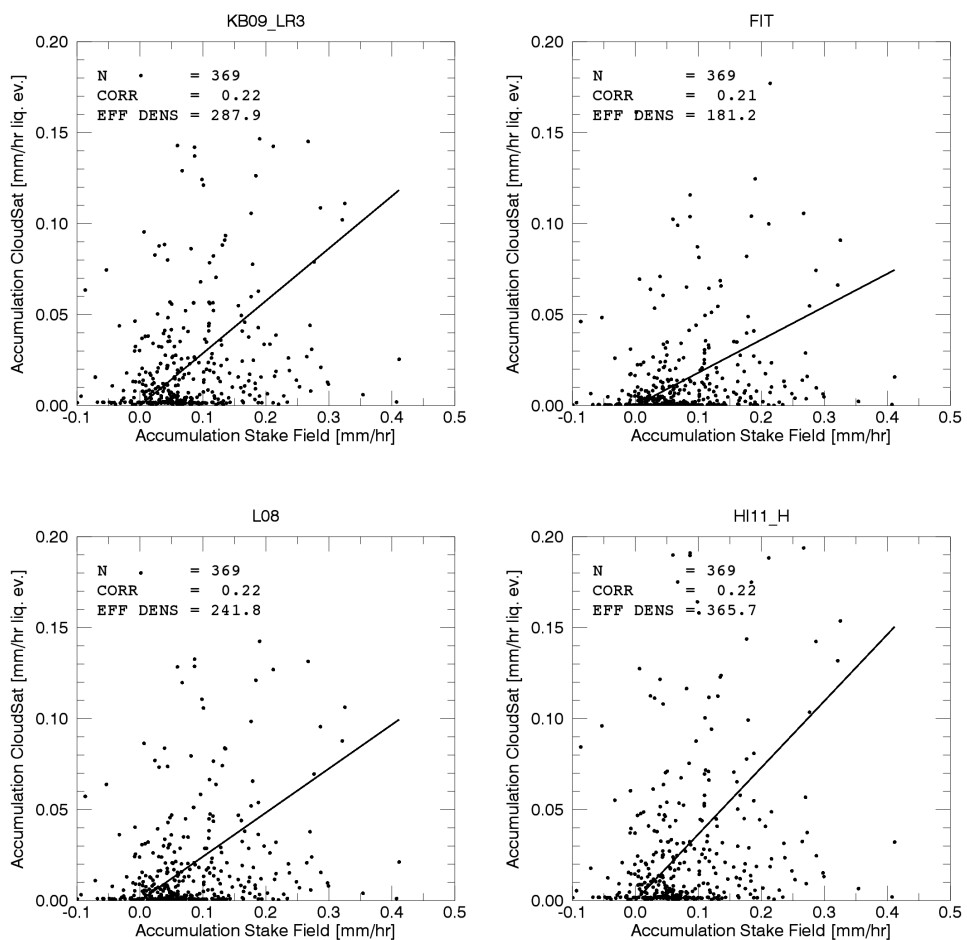

Figure 12: Same as Figure 11 but for CloudSat versus stake field.





Figure 13 shows the accumulation at Summit between 2007 and 2016 based on 369 weeks where concurrent CloudSat observations were available, and using the effective densities reported in Figure 12 for the three different Z-S relationships. Total accumulation based on these estimates is about 65 cm/yr, derived from the 4.75 meters of accumulation seen in Figure 13 over the 369 weeks.

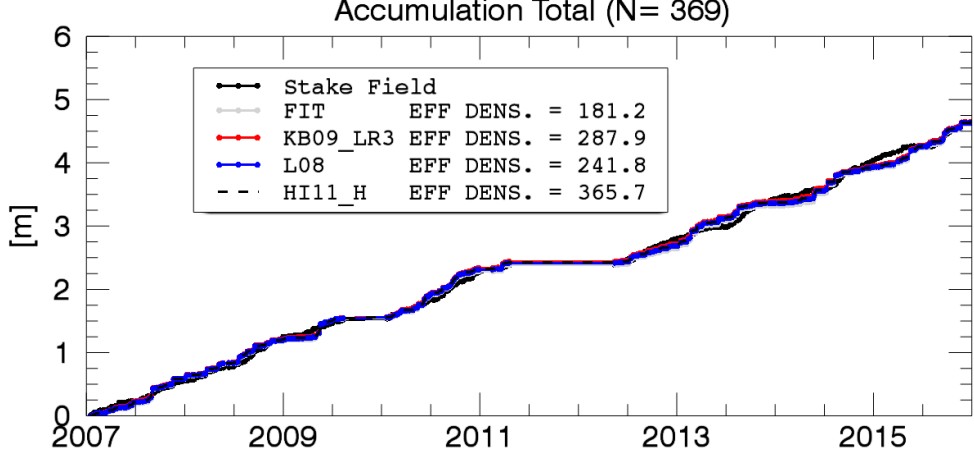

**Figure 13: Total snowfall from CloudSat and stake field for all 369 weeks where data was available for both CloudSat and the stake field.**

Based on the findings presented here, we will from here on apply the three Z-S relationships that produced realistic effective densities and average them to obtain a final surface snowfall estimate for each CloudSat observation. The spread between the three relationships will be used to determine an uncertainty range.

## 3.5 Final form of retrieval used

Based on the findings in the previous sections, the final CloudSat processing used from here on consists of three steps:

(1) We correct for topography issues and identify the lowest radar bin above the surface not affected by ground clutter using the IceBridge BedMachine topography as outlined in Section 3.1. This step results in a set of radar reflectivities observed by CloudSat typically at altitudes around 1200 m above the surface.

(2) We correct the so-obtained reflectivities for the height difference between their observation height (around 1200 m) and the surface following the method outlined in Section 3.2. This step results in a set of height-corrected reflectivities.




(3) We then apply the three Z-S relations ('HI11_H', 'KB09_LR3', and 'L08', see Section 3.3) to convert those reflectivities to equivalent snowfall rates. We average the three estimates to get a final surface snowfall estimate and use the spread between the three as an estimate for uncertainties related to the choice of Z-S relationship. Throughout the entire process we use the official CloudSat SNOWPROF product solely to determine precipitation type. That is, only if the SNOWPROF product reports snowfall, we use the snowfall rate derived according to the approach outlined here. This screens out cases with high reflectivity that are associated with rainfall.

These so-derived snowfall rates will form the basis of all further discussion from here on.

## 4 Comparison of CloudSat with ERA-Interim snowfall climatology

Using the strategy laid out in the previous section, we derive monthly mean CloudSat estimates over the GrIS for all months, where CloudSat data are available, and at a resolution of 1×2 degrees, which roughly corresponds to 111 × 111 km at 60° N. In comparison, the resolution of ERA-Interim reanalysis at 60° N is 0.7 × 0.7 degrees or 78 × 39 km. Figure 14 shows the annual mean values for CloudSat (left) and ERA (middle), as well as their relative difference in percentage (right). Figure 15 shows the annual cycle over the GrIS.

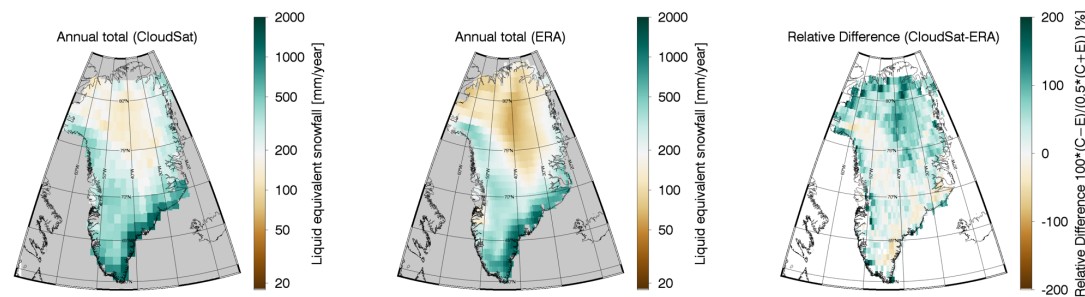

**Figure 14: Annual mean liquid equivalent snowfall from CloudSat (left panel), ERA-Interim (2006-2016, middle panel), and the relative difference between both (right panel).**

Marked differences between ERA-Interim reanalysis and CloudSat exist in the months June-September, where ERA shows less precipitation over the GrIS than CloudSat. For the summer months, the spatial correlation between ERA and CloudSat is also worst. Differences are most pronounced over the high GrIS north of 72° N, where ERA shows very little precipitation. These differences can also be identified in the monthly snowfall plots shown in Figure 16 and Figure 17. It is interesting to note that the area where the ERA-Interim reanalysis product seems to underestimate snowfall coincides nearly perfectly with areas where the CloudSat-derived snowfall is associated with low, cumuliform snowfall (see Figure 10a in Kulie et al. (2016)). The months with the highest positive bias (Figure 15, bottom left panel) also show the lowest spatial correlation between ERA-Interim reanalysis data and CloudSat snowfall estimates (Figure 15, top right panel).





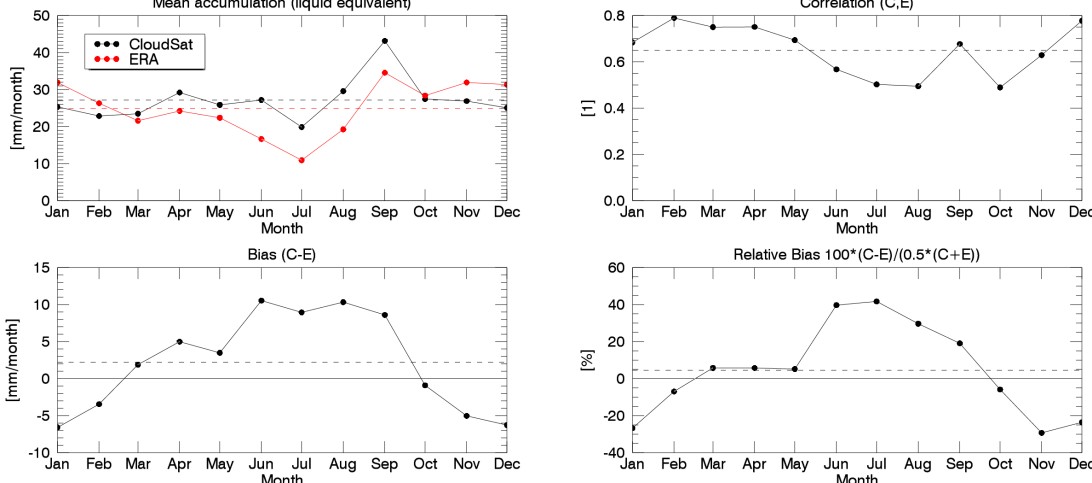

**Figure 15: Annual cycle of liquid equivalent snowfall over the GriS from CloudSat and ERA-Interim (top left panel), spatial correlation between the two (top right), mean bias (bottom left) and mean relative bias (bottom left). The dashed horizontal lines represent the annual average.**

Total snowfall over the GrIS from CloudSat adds up to $34 \pm 7.5$ cm/yr liquid equivalent, where the uncertainty range is given by the spread in Z-S relationships. The ERA estimate is 30 cm/yr. Comparing these results to an earlier publication (see Table 1 in Cullather et al. (2014)), we find our ERA-Interim reanalysis estimate to be lower. The various total snowfall values reported in Table 1 in Cullather et al. (2014) show a wide spread depending on which model was used. Further, the values in

Cullather et al. (2014) refer to total precipitation, whereas our values are snowfall only. Ettema et al. (2009) find a fraction of 6 % liquid and 94% snow over the GrIS, which can only partly explain the bias we see for ERA-Interim compared to Cullather et al. (2014). Snowfall rates from CloudSat are in better agreement with other studies. For example, Ettema et al. (2009) report snowfall over the GrIS based on high-resolution model simulations to be 40.7 cm/year (94% of their total precipitation), which is higher than both the CloudSat and ERA estimates reported here, but still in agreement with CloudSat within the range of

uncertainty.

Figures 16 and 17 show the monthly mean spatial distribution of snowfall from CloudSat (Figure 16) and ERA (Figure 17). Both datasets identify a band near the southwest coast of Greenland, where snowfall in summer is near zero. These coastal areas are presumably too warm in summer for snow to reach the ground before melting. Note that the CloudSat data are at a coarser (1x2 degree) resolution than the ERA-Interim reanalysis data, which explains this narrow coastal feature to be less

pronounced in CloudSat compared to ERA.



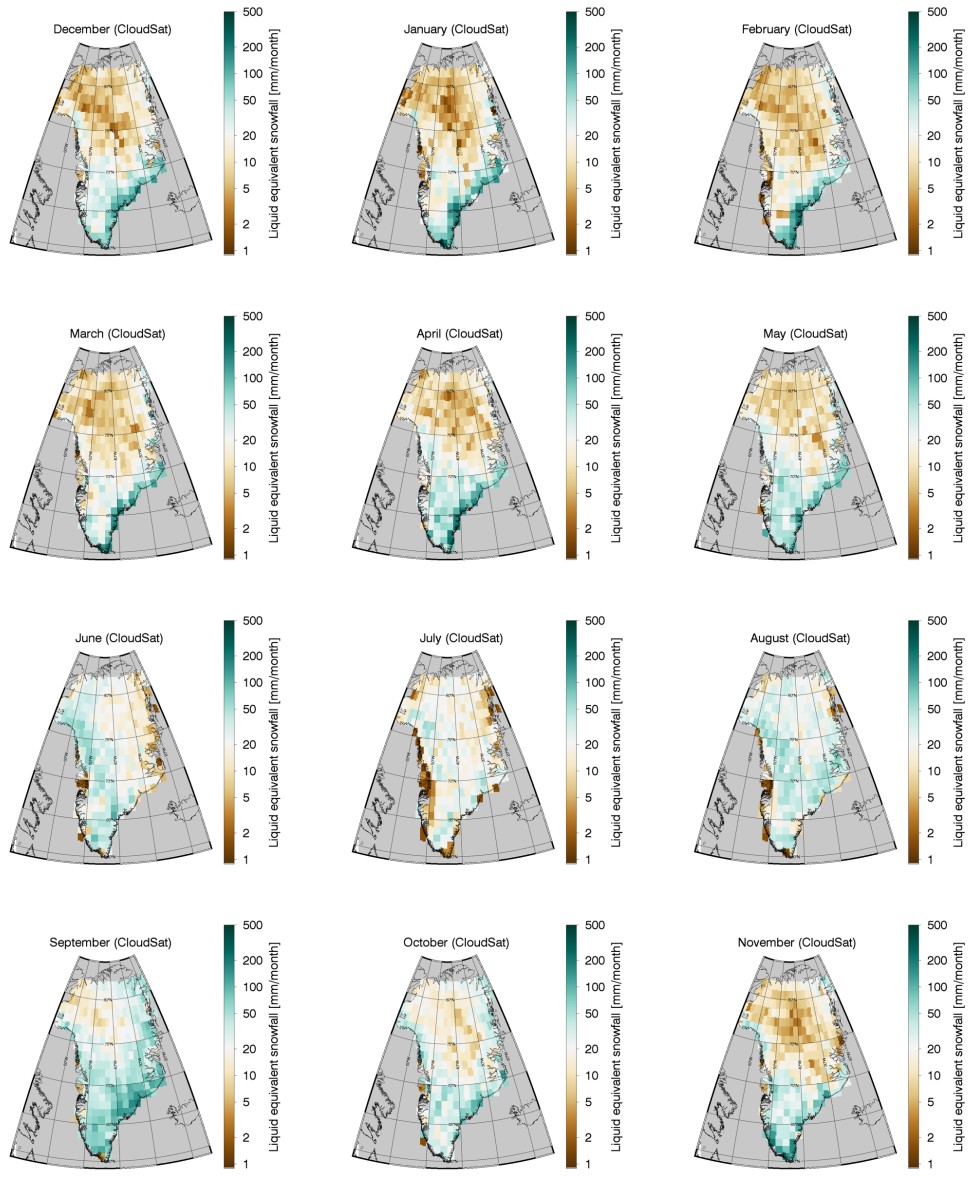

Figure 16: CloudSat-derived monthly mean snowfall rates.



**Figure 17: Same as Figure 16 but for ERA-Interim.**





### 4.1 Snowfall by drainage system

As can be seen in Figure 14, even at a resolution 1 × 2 degrees, the monthly CloudSat precipitation estimates over the GrIS are relatively noisy. As a nadir-looking instrument, CloudSat only provides few overpasses per grid-box per month. In addition to grid-box-averaged precipitation estimates, we therefore also evaluate CloudSat per major GrIS drainage system. We first
5    binned CloudSat data onto the 0.7° × 0.7° ERA-Interim reanalysis grid and subsequently averaged these gridded data onto the drainage basins.

Figure 18 and Figure 19 show the annual cycle of precipitation for the different major GrIS drainage areas as defined by Zwally et al. (2012). Consistent with earlier studies (Berdahl et al., 2018), the south-east of Greenland experiences the highest mean snowfall, and snowfall there peaks typically in wintertime. We note that the snowfall values reported in Berdahl et al. (2018)
10    are much higher than our estimates, but also higher than other published estimates (e.g. Cullather et al. (2014)). This appears to be related to Berdahl et al.'s use of only coastal stations, which experience more precipitation than inland (M. Berdahl, pers. comm. 10 May 2018). In contrast, much of the northern parts of the GrIS receive very little snowfall, but peak snowfall in those areas is in August. These features can also be observed in Figures 16 and 17.





**Figure 18:** Annual snowfall associated with the different drainage systems defined by (Zwally et al., 2012). The upper left plot shows the mean annual snowfall, the upper right plot shows the fractional contribution to the total snowfall over the GrIS, the lower left plot shows the month of maximum precipitation as well as the drainage-basin identifier used in (Zwally et al., 2012), and the lower right plot shows the amplitude of the annual cycle of snowfall. The month of maximum precipitation and amplitude are derived using a cosine fit to the annual cycle (see Figure 19).





Figure 19 compares the annual cycle of snowfall between CloudSat and ERA for all drainage areas. With few exceptions, the annual cycles between CloudSat and ERA are very similar. Furthermore, the summertime negative bias of ERA is apparent for many of the more northern drainage areas (e.g. Area 1.1). In some areas on the east coast of Greenland (e.g. Area 3.3), the agreement between CloudSat and ERA is strikingly good. The agreement in these areas seems to indicate that snowfall

5   associated with cyclonic activity over the southeast of Greenland is represented well by ERA, whereas snowfall potentially associated with summertime precipitation is potentially under-represented in the reanalysis model.

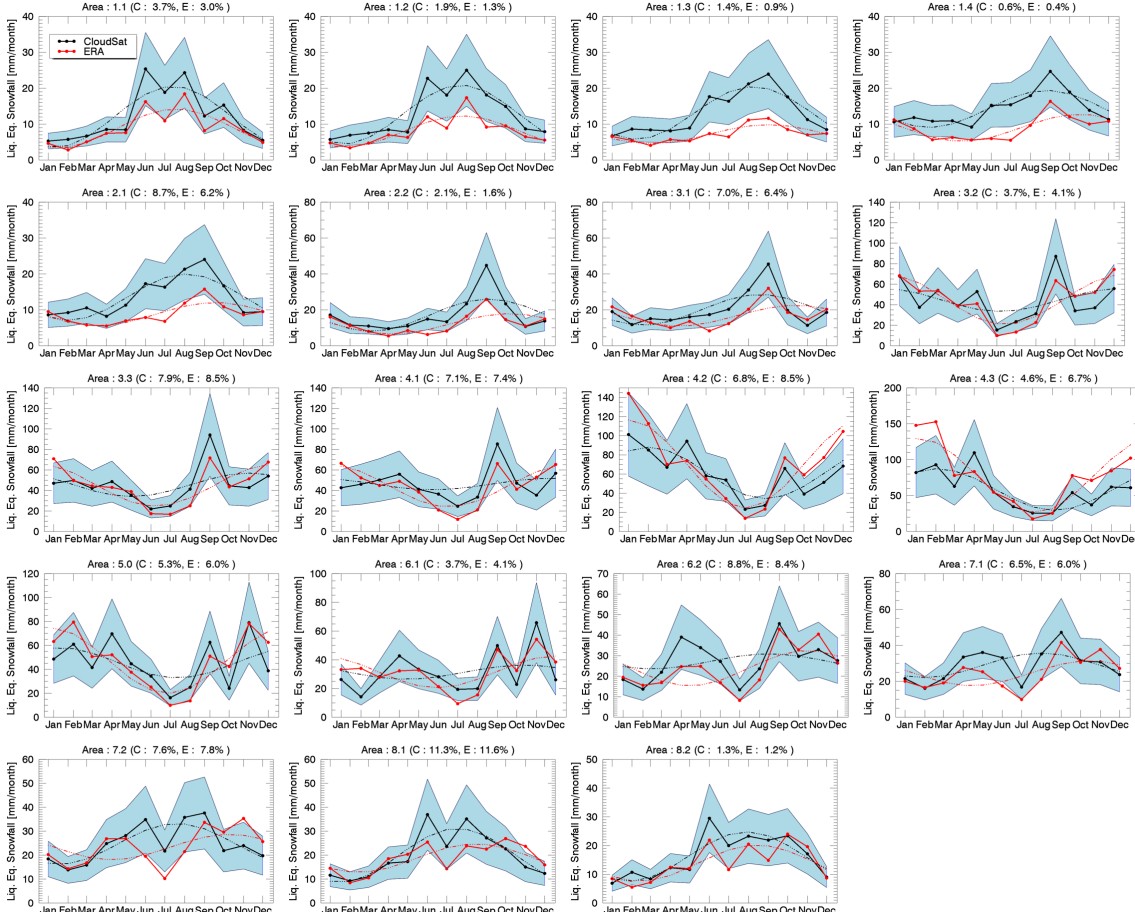

**Figure 19: Annual cycle of precipitation for the different drainage systems shown in Figure 18 for both CloudSat and ERA-Interim (2006-2016). The shaded area gives the uncertainty due to Z-S relation. The dashed curves give a cosine fit to the annual cycle. The numbers in brackets in the total are the factional contribution of each drainage area to the total snowfall over the GrIS. Note the different scales of the y-axes.**



The dashed curves in Figure 19 are cosine fits of the annual cycle of precipitation, which are used to determine the months of maximum snowfall as well as the amplitude of snowfall reported in Figure 18. We note that these cosine fits do not necessarily correspond to physical features in the annual cycle of precipitation for all drainage systems, so the values given for the annual cycle in Figure 18 should not be interpreted too quantitatively. It does appear, however, that large parts of the central and

north-western GrIS see maximum precipitation in summer, whereas the south-eastern part of the GrIS sees maximum precipitation in winter.

In the next section we further investigate the differences between ERA-Interim reanalysis and CloudSat based on monthly mean snowfall accumulation over Summit, where independent observational data are available.

**4.2 Annual cycle of snowfall at Summit**

Figure 20 compares monthly mean snowfall rates over Summit from all data sources discussed here. The stake field data have been corrected for sublimation/deposition using ERA estimates, and converted to liquid equivalent snowfall in order to make results directly comparable to the other snowfall estimates. Cullen et al. (2014) study sublimation/deposition over the high GrIS in detail and find the contribution of sublimation/deposition to be generally around 2% of the total accumulation. On a monthly basis, values from ERA were a bit higher, but still did not significantly alter the stake field values. The CloudSat

snowfall estimates agree well with the stake field with the exception of June and July, where CloudSat (as well as the MMCR) report much higher snowfall than the stake field. A similar discrepancy between June/July stake field observations and other snowfall measurements was already reported by Dibb and Fahnestock (2004). Castellani et al. (2015), in their Figure 4, show a similar behaviour. Notably, June and July are the months with the highest inter-annual variability in snowfall. For completeness, we have also included the annual cycle based on the original CloudSat SNOWPROF retrieval (green). One can

see that using the SNOWPROF surface snowfall rate retrieval without the corrections discussed and applied here would yield a precipitation estimate lower than ERA, and would also fail to show the strong annual cycle seen in the other observational datasets.

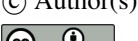



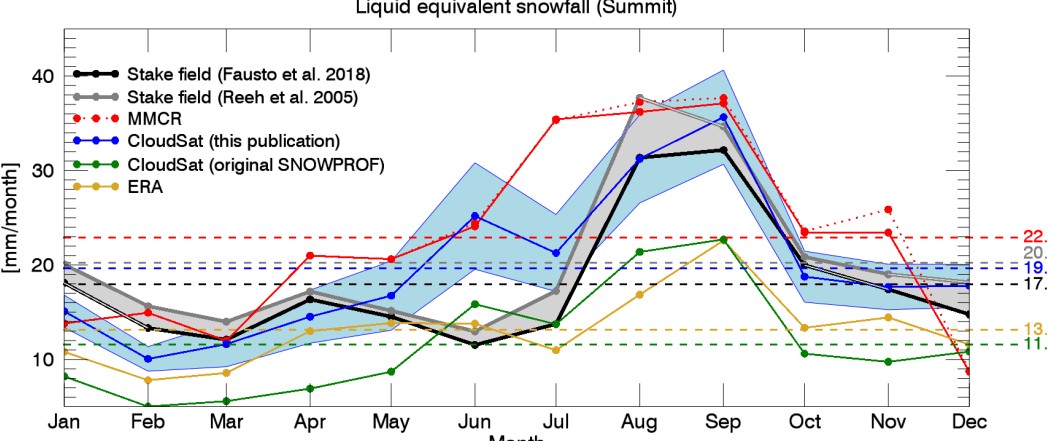

**Figure 20: Liquid equivalent snowfall over Summit from different data sources used in this publication. The shaded uncertainty range around the CloudSat estimate gives the uncertainty due to Z-S relation. The shaded uncertainty range around the stake field estimates gives the uncertainty due to different assumption about snow density. The stake field estimates presented herein have been corrected for sublimation/deposition using ERA-Interim sublimation/deposition estimates. For MMCR we used the KB09_LR3 Z-S relation, see Table 1.**

## Conclusions

Snowfall estimates from space-borne radars such as CloudSat provide estimates of snowfall in remote regions where networks of ground-based observations are difficult to establish. At the same time, these space-borne observations are difficult to validate

and potential systematic error sources related to ground clutter and the height of the space-borne observations over ground need to be addressed. Here we provided simple approaches to mitigate these adverse effects for the GrIS. We rely on auxiliary information, namely on the IceBridge topography, to better characterise surface height, and on the MMCR as a proxy for understanding the relationship between radar reflectivity at the actual observation height of CloudSat and near the surface. In particular, the correction for the observation-height based on MMCR is strictly empirical and can certainly be improved upon

for example by more explicitly accounting for physical processes, e.g. sublimation, between the height of observation and the surface. This issue is not unique to the GrIS, but might affect CloudSat snowfall estimates in other parts of the world as well. Despite remaining uncertainties, our assessment of CloudSat-derived snowfall over the GrIS shows in general good agreement with more direct ground-based observations at Summit. Since CloudSat is a nadir-only radar, its sampling does not allow for a fine horizontal resolution, and even at a resolution of 1x2 degrees CloudSat estimates of annual mean snowfall show

considerable noise. We have therefore sampled CloudSat snowfall estimates into larger areas that coincide with Greenland



drainage basins. Comparing the annual cycle of CloudSat-based snowfall with ERA-Interim for these areas shows that the ERA-Interim reanalysis generally produces robust results in areas where snowfall appears dominated by the large-scale precipitation systems. On top of the GrIS and in more northerly areas, ERA-Interim appears to miss some of the summertime precipitation that is related to shallow, more convective precipitation. This result might well be specific to ERA-Interim

reanalysis, and in future studies it will be interesting to test different climate models and reanalyses against the CloudSat dataset.

The Summit observatory plays a critical role in evaluating space-based observations of precipitation over the GrIS. It provides independent observations of weekly snowfall accumulation via a bamboo stake field; these data go back to 2007. Furthermore, the ICECAPS experiment provides upward-looking radar observations of precipitation going back to 2010, with a likely end

date of summer 2020. The validation results in this report rely heavily on this unique dataset. In order to validate space-based snowfall estimates over the GrIS, at least one reference station over the higher GrIS should be maintained and should provide routine observations of precipitation with an upward-looking radar accompanied by regular ground-based accumulation observations at not too long temporal intervals (e.g. weekly). Currently, Summit station is well equipped to fill this role. It is unclear if after summer 2020 radar observations will remain available at Summit.

Uncertainty in Z-S relationships caused by unknown and variable microphysical properties of snowfall plays an important role over the GrIS as well as elsewhere. Over the higher GrIS in particular, snowfall appears dominated by smaller ice crystals with little riming. Consequently, we find that Z-S relationships built on such particles compare better to surface observations at Summit than other Z-S relationships that were devised with mid-latitude snowfall in mind. Here, we apply the Z-S relationships that provide a reasonable fit at Summit to the entire GrIS, which might potentially lead to biases in lower-laying regions.

Further study of microphysical properties of snowfall over these regions might help address these uncertainties. Existing networks of automated weather stations such as PROMICE or GC-NET already provide limited information on snow accumulation, but could be expanded to include more direct precipitation observations.

Given the validation results reported here, we are confident that space-based radar observations from CloudSat and, in the future, EarthCare (Illingworth et al., 2015) will allow for mapping of surface precipitation over the GrIS at an accuracy high

enough to validate large-scale accumulation at seasonal temporal scales and spatial scales of the size of individual drainage basins. The nadir-only observation geometry of CloudSat and EarthCare hampers evaluation at finer spatial scales. Future space-based precipitation missions with scanning radars might provide even more information about snowfall over the GrIS. However, minimum detection thresholds need to be low enough for such instruments, as 50 % of the precipitation falling over the high GrIS is associated with reflectivities between -10 dBz and +5 dBz. Another important factor is that the range resolution

of future radars needs to be carefully tuned so as to avoid ground clutter issues, in particular when shallow precipitation is dominant. With the above caveats, space-based radars can also help provide validation datasets for other missions, such as radar or lidar altimeter missions. If integrated over large enough temporal and spatial scales, the radar-derived precipitation accumulation adjusted for sublimation/deposition should equal surface height change observed by altimeter to within the





accuracy of the snowpack density needed to covert from liquid equivalent precipitation to geometric surface height change from accumulation.

## Author Contributions

RB, FF, and DS designed and conceptualized the study. RB performed the investigation and did all of the data analysis. CP
and MS helped analyse and interpret the ICECAPS data. RB wrote the original draft. All co-authors contributed to reviewing and editing this draft.

## Acknowledgements

RB, CP, and MS were supported by NSF grants 1304544, 1314127, and 1314156. Informus GmbH was supported through ESA's SNOWSUM project. CloudSat observations were obtained from the CloudSat Data Center. ERA-Interim data were
obtained from UCAR.

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
