# Peer review of "Spatial and temporal variability of snowfall over Greenland from CloudSat observations"

_Atmospheric Chemistry and Physics, 2018_

## Referee Comment (RC1) · Anonymous Referee #1 · 26 Jan 2019

General comments :

The paper presents a method for assessing the snowfall rate from CloudSat over the Greenland ice sheet. While a standard CloudSat product (2C-SNOW-PROFILE product) provides snowfall estimations over Greenland, it has been shown that this product is affected by ground clutter contamination over some areas of the Greenland ice sheet. In this study, the authors present an interesting approach for removing ground clutter contamination by using another digital elevation model that the one used in the Cloud-Sat 2C-SNOW-PROFILE product. Moreover, an empirical correction is also performed in order to account for precipitation processes occurring between the height of the observed CloudSat reflectivities and the surface. After correcting the reflectivity profiles from CloudSat, the authors have applied three Z-S relations to assess the snowfall rate

from CloudSat. Then, the snowfall climatology derived from CloudSat has been compared to ERA Interim reanalysis in the last section of the paper. I think that the method described in this paper is interesting and represents a significant improvement in the assessment of Greenland snowfall. However, I have some comments that should be addressed before publication of the manuscript. Please find these comments below:

"Accumulation" is often used for describing the ice sheet surface mass balance. As snowfall is only one component of the surface mass balance, it would be better to use another word for the snowfall derived from ground-based and spaceborne radars such as "accumulated snowfall" or just "snowfall".

Section 3.1: A recent paper has pointed out some ground clutter issues over Greenland in the CloudSat 2C-SNOW-PROFILE product:

Palerme, C., Claud, C., Wood, N. B., L'Ecuyer, T., & Genthon, C. (2018). How does ground clutter affect CloudSat snowfall retrievals over ice sheets?. IEEE Geoscience and Remote Sensing Letters. doi:10.1109/LGRS.2018.2875007

Section 3.2: It would be interesting to see a map showing the effect of the ad-hoc correction (equation 1) on the Greenland snowfall rate. What is the spatial variability of the ad-hoc correction in terms of snowfall rate ?

Section 3.4: The authors did not discuss the possible effect of blowing snow on the snow accumulation measured by the stake field. Would it possible that blowing snow significantly affects the snow accumulation measured by the stake field ?

Specific comments :

P3, line 23: I would not say "leading to a slight annual cycle in the number of observations available over Greenland" since the number of CloudSat observations is reduced by about 50 % during the winter.

P5, figure 2, caption: "CloudSat data density over Greenland at 1x2 degrees (left, approximately 100 km x 100 km) and 0.025 x 0.05 degrees (right, approximately 2.5

km x 2.5 km)."

Please provide the latitude at which "1x2 degrees" is equal to 100 km x 100 km and 0.025 x 0.05 is equal to 2.5 km x 2.5 km.

P13, lines 18-19: I don't understand the following example: "For example, for an observed reflectivity of -30 dBz, the correction is +4 dB leading to a corrected reflectivity of -26 dBz, which still does not produce any significant snowfall".

From equation (1): dBz corrected = dBz 1000...1500 + [ (1− 0.2 * dBz 1000...1500 ) > 0 ],

For -30 dBz: dBz corrected = -30 + 1 − 0.2*(-30)

dBz corrected = -23

P20, line 7-8: "We correct for topography issues and identify the lowest radar bin above the surface not affected by ground clutter using the IceBridge BedMachine topography as outlined in Section 3.1"

I would not say "and identify the lowest radar bin above the surface not affected by ground clutter" but rather that the 5th bin above the surface deduced from the IceBridge BedMachine topography is used for estimating the surface snowfall rate. There is no test performed on the vertical profiles of reflectivity in order to identify ground clutter contamination.

P22, line 1: How is the mean snowfall rate from CloudSat calculated ? Is it calculated from the monthly averages in order to avoid the sampling issues ? This should be reported in the text.

P30, lines 4-6: "This result might well be specific to ERA-Interim and in the future studies it will be interesting to test different climate models and reanalysis against the CloudSat observations."

Behrangi et al., 2016 have compared Greenland precipitation from CloudSat to several

reanalyses (ERA-Interim, MERRA, and NCEP).

Behrangi, A., Christensen, M., Richardson, M., Lebsock, M., Stephens, G., Huffman, G.J., Bolvin, D., Adler, R.F., Gardner, A., Lambrigtsen, B., & Fetzer E. (2016) Status of high-latitude precipitation estimates from observations and reanalyses. J. Geophys. Res.Atmos. 121, 4468–4486.
* * *

---

## Referee Comment (RC2) · Lemonnier (Referee) · 31 Jan 2019

Dear Dr. Jayanarayanan Kuttippurath,

I have reviewed the manuscript "Spatial and temporal variability of snowfall over Greenland from CloudSat observations" by Ralf Bennartz and colleagues, submitted for publication in Atmospheric Chemistry and Physics.

The paper explores various datasets in order to assess the accumulation of snow over Greenland. The authors begin by evaluating the ground clutter of the Cloudsat satellite, comparing it with altimetry measurement, thus interfering with the signal of the snowfall rate. They then correct this radar signal to compare it with a ground radar (MMCR from ICECAPS) to determine a correction for the precipitation process between about

1200 and the ground level. Finally, they propose a set of Z-S relationships applied to CloudSat based on Summit observations to determine a snowfall rate on the surface, compared to snow accumulation observations by stacking method.

The authors conclude their study with an estimate of snow accumulation above Greenland of $34 \pm 7.5$ cm/year, with an accumulation above Summit of $23 \pm 4.5$ cm/year, consistent with surface observations where ERA-Interim under-estimate this rate.

The topic of the paper is certainly appropriate for Atmospheric Chemistry and Physics, and assesses the CloudSat measurements as an effective tool for studying Greenland precipitation. I have already proposed several questions and suggestions about this paper, the quality of the writing has indeed been greatly improved, but most of my questions remain unresolved.

Sincerely,

Florentin Lemonnier

**Science questions**

- Page 8, line 20 - When you re-derive the snowfall rate based on the $5^{th}$ bin of CloudSat, could you explain how do you proceed ? For example, if the IceBridge BedMachine surface topography bin is lower than the CloudSat digital elevation model, are you considering a linear interpolation of the difference of both topographies between the $4^{th}$ and the $5^{th}$ bins of CloudSat?

- Page 8, line 24 - It is indicated that differences between the two digital elevation models are caused by melting in ablation zones. In the context of global warming, using such a long dataset as CloudSat without comparing it to a topographic dataset that does not evolve over time could it add uncertainties?

- Figure 3 - As in this figure, could you show a figure of the initial and corrected

snowfall rates? I suppose that the ground-clutter affected measurements, as seen in figure 4, could be observed when the differences in elevation between BedMachine and CloudSat are strong.

- Page 10, line 11 - How did you establish this equation?

- Figure 7 - This correction seems to overestimate the precipitation below 5 dBZ and underestimate the precipitation values between 5 dBz and 20 dBz, as seen in the right panel. In other studies, such as Souverijns et al., 2018, a difference between measurements at 1200 (CloudSat reference level) and 300 (Micro-rain radar first bin) was calculated in order to propose a discrepancy between 1200 m and surface snowfall rates, what could it provide for this study?

- Figure 10 - Again, the correction you are applying on the different Z-S relationships seems to overestimate surface precipitation. A first comparison between CloudSat and MMCR datasets, then would a comparison between MMCR 1200 m.a.g.l. and ground levels be more interesting for evaluating the rate of snowfall at the surface?

- Figure 10 - What are the confidence intervals of these Z-S relationships?

- Section 3.4.2 - In my opinion, this part should be further developed. Indeed the revisit time at Summit is low and the satellite must certainly be missing a large number of events, as seen on figure 12. However, it seems to overestimate other events significantly. Could you first study and show the occurrence of Summit precipitation events observed by ground instruments, stacking field and satellite, thus recomparing the observed ground precipitation rates with the assumed CloudSat rates? As done in the study by Souverijns et al., 2018, could you discuss about commissions and omissions errors of the satellite in order to give a better estimate of the precipitation rate spatial and temporal uncertainties of

CloudSat? Are you taking into account and can you quantify blowing snow processes over stacking field on the accumulation rate, that are impossible to take into account on CloudSat measurements, occurring at low altitude?

- Page 22, line 1 - You are assuming an uncertainty on CloudSat measurement over the GrIS of 7,5 cm/year based on the spread of the Z-S relations, but hat confidence do you place in these relationships?

- Figure 19 - What does this study show about the period from 2006 to 2011, before the failure of CloudSat, where day and night observations can be considered?

- Figure 20 - Are these CloudSat measurements assumed at the surface or observed at 1200 m.a.g.l.? What is the S-Z relation used for the MMCR?

**References**

Souverijns, N., Gossart, A., Lhermitte, S., Gorodetskaya, I. V., Grazioli, J., Berne, A., Duran-Alarcon, C., Boudevillain, B., Genthon, C., Scarchilli, C., and van Lipzig, N. P. M.: Evaluation of the CloudSat surface snowfall product over Antarctica using ground-based precipitation radars, The Cryosphere Discussions, 2018, 1–21, https://doi.org/10.5194/tc-2018-111, https://www.the-cryosphere-discuss.net/tc-2018-111/, 2018b.

---

## Referee Comment (RC3) · Anonymous Referee #3 · 3 Feb 2019

This paper aim to improve the CloudSat-derived snowfall over the GrIS by removing the ground clutters, if any. Further, snowfall rate from CloudSat and ERA-Interim data are compared over the GrIS region. Trend of snowfall rate is also discussed. The results are interesting and suitable for publication in ACP after some revisions.

P2, Ln 12: van den Broeke et al. (2016) point . . . This is a complex sentence. Please rewrite.

P3, Ln 13: . . .however, only daytime scene. . . The author considered daytime data only after 2011, while both daytime and night-time data are considered between 2006 and 2011. Is there any bias in the results due to diurnal data?

P6, Ln 19: To compare CloudSat CPR and the MMCR,. . .. It is not clear what is

compared, reflectivity or snowfall rate.

P6, Ln 24: . . . allow transferring results between the two bands. The sensitivity of CloudSat CPR and MMCR may be different. What is the sensitivity of MMCR. In comparison analysis, do the authors considered sensitivity of both radars. Ln32: (2011))0. Check it.

P8, Ln 20: re-derive the snowfall rates based on the 5th radar bin above the surface. Any specific reason for choosing 5th bin .

P 9, Ln 5: . . ..significant number of high reflectivities associated with very high. . . Any role of Mie-scattering?

P12, Ln14: Vertical resolution of CloudSat is 240 m as mentioned in Page 3.

P12, Ln 15: How the equivalent radar reflectivity measured at W-band compared with Ka-band measurements? Sensitivities of both the radars are different and thus the comparison of reflectivity is not so straightforward. The authors mentioned about the spatial averaging but didn't mention about the temporal averaging for performing the radar reflectivity comparison?

P13: How eq(1) is derived?

P13: Ln 18: -20 dBz. I couldn't see -30 dBz in the figure 7.

P17: Ln 6:. . .. effective density. . . Please elaborate "effective density". If effective density is higher what it mean?
* * *

---

## Author Comment (AC1) · 15 Apr 2019

We thank all three reviewers for their thorough and constructive comments. We hope to have clarified the issues the reviewers brought up and we feel that the revised version of the manuscript is significantly improved.

Please see our detailed response to all three reviewers' comments in the attached PDF.

Please also note the supplement to this comment: https://www.atmos-chem-phys-discuss.net/acp-2018-1045/acp-2018-1045-AC1-supplement.pdf
* * *
[Figure]

2019.

---

## Author Comment (AC2) · 15 Apr 2019

We thank all three reviewers for their thorough and constructive comments. We hope to have clarified the issues the reviewers brought up and we feel that the revised version of the manuscript is significantly improved.

Below please find our detailed response to all three reviewers' comments.

**1.     Reviewer 1**

**"Accumulation" is often used for describing the ice sheet surface mass balance.  As snowfall is only one component of the surface mass balance, it would be better to use another word for the snowfall derived from ground-based and spaceborne radars such as "accumulated snowfall" or just "snowfall".**

We appreciate the distinction between actual accumulation and snowfall. In the paper we now use 'snowfall accumulation' throughout to describe the latter, while we refer to accumulation whenever we compare against or use the stake field measurements. Note, that we discuss the terms other than snowfall that contribute to the total accumulation in the paper. It is also worth noting that at least over the higher Greenland Ice Shield (GrIS), snowfall is by far the dominant term in the actual accumulation as noted toward the end of Section 3.4.1.

**Section 3.1: A recent paper has pointed out some ground clutter issues over Greenland in the CloudSat 2C-SNOW-PROFILE product: Palerme, C., Claud, C., Wood, N. B., L'Ecuyer, T., & Genthon, C. (2018).  How does ground clutter affect CloudSat snowfall retrievals over ice sheets?. IEEE Geoscience and Remote Sensing Letters. doi:10.1109/LGRS.2018.2875007**

We thank the reviewer for pointing us to this paper, which was published in December 2018, a few months after our discussion paper was finished (submitted September 2018). The results that Palerme et al. present regarding ground clutter are in very good agreement with our findings and fully support our analysis. We have added the citation.

**Section 3.2:  It would be interesting to see a map showing the effect of the ad-hoc correction (equation 1) on the Greenland snowfall rate. What is the spatial variability of the ad-hoc correction in terms of snowfall rate?**

This information is shown in Figure 6. In particular, there is a hump visible in the black curve at around 20-30 dBz. This hump is caused by ground clutter. It adds about 10% to the grand-mean snowfall rate. However, keep in mind that those increases occur only in areas with heterogeneous topography (Figure 5) and do not affect the higher GrIS.

In order to make this clearer, we have now added this discussion and point explicitly to the 20 - 30 dBz range.

**Section 3.4: The authors did not discuss the possible effect of blowing snow on the snow accumulation measured by the stake field. Would it possible that blowing snow significantly affects the snow accumulation measured by the stake field?**

This issue has been discussed in earlier publications and is generally thought to be of lesser importance. We have added a brief discussion on blowing snow toward the end of Section 2.2

**P3, line 23: I would not say "leading to a slight annual cycle in the number of observations available over Greenland" since the number of CloudSat observations is reduced by about 50 % during the winter.**

We agree and have dropped the word 'slight'.

**P5, figure 2, caption: "CloudSat data density over Greenland at 1x2 degrees (left, approximately 100 km x 100 km) and 0.025 x 0.05 degrees (right, approximately 2.5 km x 2.5 km)." Please provide the latitude at which "1x2 degrees" is equal to 100 km x 100 km and 0.025 x 0.05 is equal to 2.5 km x 2.5 km.**

Valid at 60 N. We have added this in the caption.

**P13, lines 18-19: I don't understand the following example: "For example, for an observed reflectivity of -30 dBz, the correction is +4 dB leading to a corrected reflectivity of -26 dBz, which still does not produce any significant snowfall". From equation (1): dBz corrected = dBz 1000...1500 + [ (1   0.2 * dBz 1000...1500 ) > 0 ], For -30 dBz: dBz corrected = -30 + 1 – 0.2*(-30) dBz corrected = -23**

This was a calculation error on our end. Thanks for catching this. We have corrected it correspondingly.

**P20, line 7-8: "We correct for topography issues and identify the lowest radar bin above the surface not affected by ground clutter using the IceBridge BedMachine topography as outlined in Section 3.1" I would not say "and identify the lowest radar bin above the surface not affected by ground clutter" but rather that the 5th bin above the surface deduced from the IceBridge BedMachine topography is used for estimating the surface snowfall rate. There is no test performed on the vertical profiles of reflectivity in order to identify ground clutter contamination.**

We have changed this accordingly.

**P22, line 1: How is the mean snowfall rate from CloudSat calculated? Is it calculated from the monthly averages in order to avoid the sampling issues?  This should be reported in the text.**

Correct. We have added this information.

**P30, lines 4-6: "This result might well be specific to ERA-Interim and in the future studies it will be interesting to test different climate models and reanalysis against the CloudSat observations." Behrangi et al., 2016 have compared Greenland precipitation from CloudSat to several reanalyses (ERA-Interim, MERRA, and NCEP). Behrangi, A., Christensen, M., Richardson, M., Lebsock, M., Stephens, G., Huffman, G.J., Bolvin, D., Adler, R.F., Gardner, A., Lambrigtsen, B., & Fetzer E. (2016) Status of high-latitude precipitation estimates from observations and reanalyses. J. Geophys. Res.Atmos. 121, 4468–4486.**

Thanks. We have added this citation and some discussion.

**2. Reviewer 2**

**Page 8, line 20 - When you re-derive the snowfall rate based on the 5 th bin of CloudSat, could you explain how do you proceed? For example, if the IceBridge BedMachine surface topography bin is lower than the CloudSat digital elevation model, are you considering a linear interpolation of the difference of both topographies between the 4 th and the 5 th bins of CloudSat?**

We do not perform any vertical interpolation of CloudSat data. We just take the $5^{th}$ bin above whatever the CloudSat surface bin is with respect to the topography. We believe an interpolation would under no circumstances be advisable since it basically reduces the vertical resolution.

**Page 8, line 24 - It is indicated that differences between the two digital elevation models are caused by melting in ablation zones. In the context of global warming, using such a long dataset as CloudSat without comparing it to a topographic dataset that does not evolve over time could it add uncertainties?**

We do not *indicate* in our paper that differences between the CloudSat topography and ICEBRIDGE are caused by melting. We only acknowledge melting could potentially contribute to the difference. In our view, it is much more likely that IceBridge is simply more accurate than the CloudSat topography, because IceBridge is actually dedicated to the GrIS topography. However, we have no way of proving this.

An impact of melting ice over time on the climatology of snowfall would only be possible insofar as to which CloudSat bin above the surface would be picked. We believe this is only a minor concern, given the uncertainties in choice of Z-S relationship and other sources of error.

**Figure 3 - As in this figure, could you show a figure of the initial and corrected snowfall rates? I suppose that the ground-clutter affected measurements, as seen in figure 4, could be observed when the differences in elevation between BedMachine and CloudSat are strong.**

We show this already at the very end in Figure 20.

**Page 10, line 11 - How did you establish this equation?**

The correction was created by fitting a regression line through the data between -20 dBz and +5 dBz. We have added that piece of information now.

**Figure 7 - This correction seems to overestimate the precipitation below 5 dBZ and underestimate the precipitation values between 5 dBz and 20 dBz, as seen in the right panel. In other studies, such as Souverijns et al., 2018, a difference between measurements at 1200 (CloudSat reference level) and 300 (Micro-rain radar first bin) was calculated in order to propose a discrepancy between 1200 m and surface snowfall rates, what could it provide for this study?**

We thank the reviewer for pointing us to the Souverijns et al., 2018 paper which was published just around the time we submitted our paper and hence we were not aware of it. As the reviewer points out, the methodology for correction employed in Souverijns et al., 2018 is slightly different than ours. It is not easily transferable to our method, as we use a set of different Z-S relationships that would all lead to different corrections if the correction was performed on S. We therefore believe that our correction, working directly on *observed* Z-values, rather than *retrieved* S-values, is more appropriate in this context. We have referenced the Souverijns et al., 2018 paper now.

It is true there is a slight overestimation of the correction but actually only up to about 0 dBz. This is compensated by a slight underestimation between 0 and +5 dBz (above +5 dBz, the correction does not apply, as stated in the text). One cannot, of course, expect a simple statistical correction to perform perfectly everywhere. The important issue is that the corrected values are much closer to the surface Z-values than the uncorrected values. This can most easily be seen by comparing the left and center panels of Figure 7.

The relative effect of the correction compared to other uncertainties can also be seen in Figure 10. The uncertainty is dominated by variability between different Z-S relationships and not by the impact of corrections.

**Figure 10 - Again, the correction you are applying on the different Z-S relationships seems to overestimate surface precipitation. A first comparison between CloudSat and MMCR datasets, then would a comparison between MMCR 1200 m.a.g.l. and ground levels be more interesting for evaluating the rate of snowfall at the surface?**

We are not sure we understand what the reviewer is asking. It is also not clear to us, where exactly in Figure 10 the reviewer sees the 'overestimation of surface precipitation'.

**Figure 10 - What are the confidence intervals of these Z-S relationships?**

We have now added a brief section on uncertainties (in Section 2.4), where we also refer the reader to literature on uncertainties of individual Z-s relationships. In the context of the current study the difference between different Z-S relationships is more important than uncertainties associated with individual Z-S relationships.

**Section 3.4.2 - In my opinion, this part should be further developed. Indeed the revisit time at Summit is low and the satellite must certainly be missing a large number of events, as seen on figure 12. However, it seems to overestimate other events significantly. […]**

This reviewer comment has three sub-questions, which we answer below.

- **[…] Could you first study and show the occurrence of Summit precipitation events observed by ground instruments, stacking field and satellite, thus recomparing the observed ground precipitation rates with the assumed CloudSat rates?**

It is not clear to us what exactly the reviewer is proposing here. However, we hope that the additional discussion we have added regarding temporal and spatial sampling does answer this question as well.

- **[…] As done in the study by Souverijns et al., 2018, could you discuss about commissions and omissions errors of the satellite in order to give a better estimate of the precipitation rate spatial and temporal uncertainties of CloudSat?**

We are not sure what the word 'commissions' means in this context. We assume the reviewer wants us to further investigate the temporal and spatial sampling associated with CloudSat. We are not sure what this additional discussion would yield. The spatial and temporal uncertainty of both CloudSat and MMCR observations in themselves do not play a role.

However, when comparing CloudSat to MMCR, the representativeness of MMCR w.r.t. to CloudSat does play a role. This representativeness depends on various issues including the temporal sampling of CloudSat, the spatial distance between CloudSat and concurrent MMCR observations, and the spatial heterogeneity of the observed snowfall scenes.

As a result, comparisons of CloudSat with MMCR will be noisier than comparisons between MMCR and other ground-based observations at Summit (e.g the stake field). The effect of this can best be studied by comparing Figures 11 (where we compared MMCR with the stake field) and Figure 12 (where we compare CloudSat with MMCR). The additional noise manifests itself in reduced correlations between CloudSat and MMCR (compared to stake field and MMCR, Figure 11).

In our paper we are concerned with retrieval issues related to systematic errors in the CloudSat data (.e.g ground-clutter, Z-S relationships). We are not as much concerned with issues related to increased noise in direct comparisons, which can be mitigated in the climatology by large-scale temporal and spatial averaging. We therefore do not see any need for extending our analysis in the way suggested by the reviewer.

- **[…] Are you taking into account and can you quantify blowing snow processes over stacking field on the accumulation rate, that are impossible to take into account on CloudSat measurements, occurring at low altitude?**

The effects of blowing snow/snow drift on the stake field as well as on accumulation over the GrIS in general have been studied earlier. See the discussion in Castellani et al. (2015). In our initial manuscript we had failed to mention these findings. We have now added a discussion of the issue toward the end of Section 2.2 addressing these issues.

**Page 22, line 1 - You are assuming an uncertainty on CloudSat measurement over the GrIS of 7,5 cm/year based on the spread of the Z-S relations, but hat confidence do you place in these relationships?**

See answer to earlier question by Reviewer 2 on 'Figure 10 – What are the…' .

**Figure 19 - What does this study show about the period from 2006 to 2011, before the failure of CloudSat, where day and night observations can be considered?**

We are not certain how to interpret the question. If the reviewer is just asking how Figure 19 looks like for the years 2006-2011, the results presented in Figure 19 do include the years 2006-2011. We do not believe the data density of CloudSat is high enough to further separate out shorter time periods.

However, it might be the reviewer is asking whether there is a bias caused by CloudSat only observing during daytime in the later time period after its battery failure in early 2012 (Reviewer #3 has that same question). This would be a more serious matter. In order to exclude this possibility, we have investigated the diurnal cycle of precipitation over Summit from our MMCR dataset, which is sampled at 1-minute intervals. The below figure (Figure R1) shows the diurnal cycle of precipitation for all four seasons. There is no diurnal cycle apparent from the MMCR, except for JJA, where the 'daytime' precipitation appears to be lower than the 'nighttime' precipitation. Note, that strictly speaking there is no nighttime at Summit in JJA, but still the solar zenith angle is lower around midnight local time. The diurnal cycle seen in Fig R1 for JJA  might or might not be a physical effect caused by this solar forcing. It would definitely be of interest to further study the potential diurnal cycle of precipitation in summertime over the GrIS. For all other seasons, no diurnal cycle is apparent in Figure R1 and this the result is consistent with expectations. During wintertime no solar forcing is expected as the sun will not rise above the horizon. During the spring and fall seasons, the solar forcing will be very weak.

In order to now answer the question as to whether the CloudSat sampling issues after 2011 could cause a bias in the observed snowfall rates, one has to keep in mind that CloudSat data in summer are NOT affected by data loss (see Figure 1). Only nighttime CloudSat observations are affected, thereby most strongly reducing the data coverage in winter, when no diurnal cycle in precipitation is present.

In summary:

It is conceivable there might be a diurnal cycle in precipitation over Summit in summertime. Figure R1 seems to indicate this is the case. However, in summertime CloudSat consistently samples the full diurnal cycle even after the battery failure in early 2012. It is therefore very unlikely, that this failure would introduce biases in CloudSat-derived monthly-mean precipitation compared to the earlier CloudSat time period before the battery failure. The reduced data amount during wintertime might lead to increased noise in the CloudSat-based snowfall estimates during the later years. Minimizing this noise is one reason, why we average the final climatology over all years available (and over larger regions).

[Figure]

Figure R1: Diurnal cycle of precipitation over Summit derived from MMCR observations. DJF: December-January-February; MAM: March-April-May; JJA: June-July-August; SON: September-October-November

**Figure 20 - Are these CloudSat measurements assumed at the surface or observed at 1200 m.a.g.l.? What is the S-Z relation used for the MMCR?**

CloudSat: observations are taken from the 5[th] radar bin above the surface and corrected for height according to the methodology we developed in the paper. For comparison, we also added the CloudSat original SNOWPROF snowfall rates. For MMCR Z-S relationship we used KB09_LR3 as a Z-S relationship (see Table 1 in the paper). We have added this information now in the figure caption.

**3.    Reviewer 3**

**P2, Ln 12: van den Broeke et al. (2016) point ... This is a complex sentence. Please rewrite.**

We have split this sentence in two now.

**P3, Ln 13: ...however, only daytime scene... The author considered daytime data only after 2011, while both daytime and night-time data are considered between 2006 and 2011. Is there any bias in the results due to diurnal data?**

Please see our detailed response (including figure R1) to Reviewer #2 who had the same question.

**P6, Ln 19:  To compare CloudSat CPR and the MMCR,....   It is not clear what is compared, reflectivity or snowfall rate.**

We have added 'snowfall rates'.

**P6, Ln 24:  ... allow transferring results between the two bands.  The sensitivity of CloudSat CPR and MMCR may be different. What is the sensitivity of MMCR. In comparison analysis, do the authors considered sensitivity of both radars. Ln32: (2011))0. Check it.**

Both instruments have a low enough sensitivity threshold (<= - 30 dBz) so that differences in their sensitivity do not make any difference in terms of retrieved snowfall rates.

**P8, Ln 20: re-derive the snowfall rates based on the 5th radar bin above the surface. Any specific reason for choosing 5th bin .**

Excellent point. This is discussed later in the manuscript but appears very ad-hoc here. In order to clarify why we chose the 5th bin, we have now added the corresponding reference.

**P 9, Ln 5: ....significant number of high reflectivities associated with very high... Any role of Mie-scattering?**

We believe Mie scattering does not play a role in this. Firstly, the actual snowfall rates over the GrIS are quite small. Secondly, Mie scattering would reduce the reflectivity observed for a given snowfall rate (as the backscatter efficiency as function of particle diameter flattens out compared to the Rayleigh regime). Also, from our analysis it is clear that we are dealing with surface clutter.

**P12, Ln14: Vertical resolution of CloudSat is 240 m as mentioned in Page 3.**

We believe the discussion in the paper regarding CloudSat's vertical resolution and sampling is correct. The CloudSat data is oversampled at a vertical resolution of 240 m. However, the actual vertical resolution of CloudSat is 500 m (as correctly stated on Page 3). That is, if a data bin is at, say 1250 m, it is representative for a range between 1000 and 1500 m.

**P12, Ln 15: How the equivalent radar reflectivity measured at W-band compared with Ka-band measurements?  Sensitivities of both the radars are different and thus the comparison of reflectivity is not so straightforward. The authors mentioned about the spatial averaging but didn't mention about the temporal averaging for performing the radar reflectivity comparison?**

The comparison of reflectivity between Ka and W-band would only be not straight-forward, if either (a) Mie-effects affects the observations or (b) one of the radars has a significantly worse detectability threshold than the other radar. The effects of (a) Mie-scattering are discussed in detail in the paper (Figure 9 and associated discussion). Both MMCR and CloudSat have a low enough detectability threshold (<= -30 dBz) that (b) is not an issue.  Regarding temporal averages, we perform temporal averages only on snowfall rates (and not on reflectivities, which would be incorrect). However, Figure 9 only deals with Z-S relations, so issues of temporal averaging to do not play a role.

**P13: How eq(1) is derived?**

The correction was created by fitting a regression line through the data between -20 dBz and +5 dBz. We have added that piece of information now.

**P13: Ln 18: -20 dBz. I couldn't see -30 dBz in the figure 7.**

The -30 dBz just serve as an example in this context. We have re-formulated the corresponding paragraph to make this clearer.

**P17:  Ln 6:....  effective density... Please elaborate "effective density".  If effective density is higher what it mean?**

We have added some explanation and a reference to Section 2.5, where effective density is discussed.

---

## Author Response (AR2)

We thank the reviewer for the additional comments. We have fixed most of them according to the reviewer's suggestions. In a view instances, we have kept our original formulation. See our detailed response to the reviewer's comments in bold below.

P01, L25: delete "so-" **FIXED**
P03, L14: The CPR - **FIXED**
P05, L09: length or volume. Are you talking about the time series? **FIXED. Replaced by 'time series'**
P6, L19: it assimilates radiance data from a number of satellites **FIXED**
P6, L20: The ERA-Interim. **We believe the way we have it is correct grammatically.**
P8, L16: the height of surface bin **We believe the way we have it is correct grammatically.**
Figures 3 and 5: there are no panels. Please write figures on the left or right. **FIXED**
P10, L03: It can be seen that **FIXED**
P10, L04: as far as the complex topography is concerned? **We believe the way we have it is correct**
P10, L14: "blind zone"; measurements are not possible or measurements cannot be used? **FIXED. We have clarified this**
P12, L01: "hereafter" **FIXED**
P12, L06-08: Just write that "Our findings are consistent with that of Palerme et al. (2019)." No need of justification regarding the discussion paper. **WE believe this is important information, so we prefer to keep this. It will allow the reader to understand that their approach and ours are developed independently.**
P13, L05: 0.3% is affected. **We believe the way we have it is correct grammatically.**
P13, L17: more correction? "an additional correction" would be a better choice **FIXED**
P13, L21: "from which it was derived" and "stem" **FIXED. WE have rewritten this sentence to clarify what we mean.**
P13, L22: "correction not to produce viable results"? What was the purpose correction here? **We believe the sentence is clear. We do not expect the correction to produce viable results outside the region it was developed for.**
P14, L16: "onset" of mie-scattering? **We believe this to be the correct use of the word onset.**
Figure 9, Caption: "… visual reference to show the difference …" **FIXED**
P15, L08: extending this analysis **It is not clear to us whwt the reviewer means.**
P17, L04: "derived" **FIXED**
P18, L09: I did not understand the sentence. Please rephrase it. **FIXED**
P18, L19: "not surprising as CloudSat provides " **FIXED**
P18, L29: I would consider your original word choice here; "effects" not impacts **FIXED**
Figure 13: It would be good to give a name for the x-axis : "Year" **FIXED**
P20, L10, 20: delete "so-" **FIXED**
P22, L10: the uncertainty of CloudSat retrievals? **FIXED**
P27, L06: "significantly underrepresented", to avoid one of the two "potentially" from the sentence. **FIXED. We got rid of one of the potentiallys**